# Interneuron diversity in the human dorsal striatum

Leonardo D. Garma [1,6], Lisbeth Harder [1,6], Juan M. Barba-Reyes[2], Sergio Marco Salas[3], Mónica Díez-Salguero[2], Mats Nilsson [3], Alberto Serrano-Pozo [4,5], Bradley T. Hyman [4,5] & Ana B. Muñoz-Manchado [1,2] ✉

Deciphering the striatal interneuron diversity is key to understanding the basal ganglia circuit and to untangling the complex neurological and psychiatric diseases affecting this brain structure. We performed snRNA-seq and spatial transcriptomics of postmortem human caudate nucleus and putamen samples to elucidate the diversity and abundance of interneuron populations and their inherent transcriptional structure in the human dorsal striatum. We propose a comprehensive taxonomy of striatal interneurons with eight main classes and fourteen subclasses, providing their full transcriptomic identity and spatial expression profile as well as additional quantitative FISH validation for specific populations. We have also delineated the correspondence of our taxonomy with previous standardized classifications and shown the main transcriptomic and class abundance differences between caudate nucleus and putamen. Notably, based on key functional genes such as ion channels and synaptic receptors, we found matching known mouse interneuron populations for the most abundant populations, the recently described PTHLH and TAC3 interneurons. Finally, we were able to integrate other published datasets with ours, supporting the generalizability of this harmonized taxonomy.

The dorsal striatum is a subcortical brain structure that in humans consists of caudate nucleus (CN) and putamen (Pu), separated by the internal capsule. Together with the ventral striatum (nucleus accumbens and olfactory tubercle), the globus pallidus, the subthalamic nucleus, and the substantia nigra, it makes up the basal ganglia nuclei[1]. The striatum carries out functions related to motor control, action learning, reward-related behavior, and cognition with certain regional preferences; the CN is mainly responsible for eye movement and cognitive functions, the Pu for motor control, learning, and auditory responses, and the ventral striatum is related to limbic functions such as reward and motivation. Dysfunction of the striatum is a key feature of neurodegenerative disorders such as Parkinson's and Huntington's diseases[2–5] as well as of psychiatric conditions such as obsessive-compulsive disorder and schizophrenia[6–8].

The dorsal striatum is the main input area of the basal ganglia and exhibits a high level of activity-dependent synaptic plasticity[9], representing a critical hub for the processing and selection of information sent to the other basal ganglia nuclei. This information is relayed through its projecting neurons, known as medium spiny neurons (MSNs) because of their morphological features[10]. MSNs, which are characterized by their inhibitory signaling via gamma-aminobutyric acid (GABA), constitute the majority of the striatal neuronal

[1]Karolinska Institutet, Laboratory of Molecular Neurobiology, Department of Medical Biochemistry and Biophysics, Stockholm, Sweden. [2]Departamento de Anatomía Patológica, Biología Celular, Histología, Historia de la Ciencia, Medicina Legal y Forense y Toxicología. Instituto de Investigación e Innovación Biomédica de Cádiz (INiBICA). University of Cádiz, Cádiz, Spain. [3]Science for Life Laboratory, Department of Biochemistry and Biophysics, Stockholm University, Stockholm, Sweden. [4]Massachusetts General Hospital, Neurology Department, Boston, Massachusetts, USA. [5]Harvard Medical School, Boston, Massachusetts, USA. [6]These authors contributed equally: Leonardo D. Garma, Lisbeth Harder. ✉e-mail: ana.munoz@uca.es

population. However, their function depends on a diverse group of locally-projecting neurons known as interneurons.

Striatal interneurons integrate incoming information from different brain areas and act on MSNs activity to modulate the output information. This filtering process is also regulated by incoming dopaminergic and serotonergic projections from the midbrain and the dorsal raphe nucleus, respectively[11,12]. Classically, striatal interneurons are categorized into six main groups based on specific markers and electrophysiological profiles[13–20]. These groups include choline O-acetyltransferase (CHAT)-expressing cells—representing the cholinergic giant neurons—and various GABAergic medium-size cells characterized by the expression of one particular calcium-binding protein such as parvalbumin (PV or PVALB), calbindin (CALB1), or calretinin (CR or CALB2); by the catecholamine synthesis rate-limiting enzyme tyrosine hydroxylase (TH); or those identified as nitrergic, i.e. expressing nitric oxide synthase (nNOS) and nicotinamide adenine dinucleotide phosphate diaphorase (NADPH-d)[21]. Nitrergic cells are also divided in those predominantly expressing both neuropeptide Y (NPY) and somatostatin (SST) and exhibiting a plateau low-threshold-spiking electrophysiological profile, and those expressing just NPY and displaying a late-spiking profile (also known as neurogliaform cells [NGCs])[22,23].

Recent advances such as new transgenic reporter mice that target the complete striatal and cortical interneuron repertoire[18,24], and single-cell/nucleus RNA-sequencing (sc/nRNA-seq) have enabled large-scale approaches to investigate cell diversity based on the individual cell transcriptome[25–27] in different mouse brain areas including the striatum[28–30]. Using these methods, a recent study identified seven interneuron populations in the mouse striatum based on their full molecular and electrophysiological profile, revealing a higher diversity than previous research suggested: Npy/Sst, Npy/Mia (identified as the NGCs), Cholecystokinin (Cck)/Vasoactive Intestinal Peptide (Vip), Cck, Chat, Th, and Parathyroid Hormone Like Hormone (Pthlh)[28]. Among them, the *Pthlh*-expressing interneurons represent a recently described class of striatal interneurons that is characterized by a variable *Pvalb* expression level and a broad continuum of intrinsic electrophysiological properties which correlates with *Pvalb* levels[28]. This continuum seems to follow a regional gradient pattern within the mouse dorsal striatum, suggesting that the different types of striatal cells receive inputs from different brain cortical areas[31]. The identification of this population elucidated the enigma of the existence of a large number of striatal 5-Hydroxytryptamine Receptor 3 A (HTR3a)-expressing interneurons that did not match any of the classical interneuron populations[18]. Nevertheless, in the human and non-human primate striatum, the majority of studies still rely on the classical classification due to the absence of a comprehensive and systematic consensus on the constituent populations within these neuronal groups[32–36]. Prior snRNA-seq studies on the human and non-human primate striatum have highlighted different aspects, such as broad differences across species and striatum vs. other brain areas[37,38] or in health vs. disease[39], but lack sufficient interneuron sampling to properly portray the diversity of striatal interneurons, leading to conflicting results and, importantly, no consensus.

In the present study, we employed snRNA-seq and spatial transcriptomics to explore the diversity of interneurons in the human dorsal striatum (CN and Pu) across 28 neuropathologically control donors. Our extensive sampling included nearly half a million nuclei overall, of which almost 20,000 were identified as interneurons. This constitutes by far the largest and more robust study of its kind to date describing the distinct striatal interneuron groups using the aforementioned highly sensitive techniques. We have leveraged this large dataset and spatial expression profile to establish the major and minor divisions between the interneuron classes and types in both regions, provide specific markers for each, and describe the differences between CN and Pu. We have also delineated an inner gradient

structure within the most abundant classes (PTHLH and Tachykinin Precursor (TAC) 3) as well as mapped their correspondence with both the previous classical human striatal taxonomy and the mouse striatal interneuron classes. Additionally, we have discovered that our taxonomy also correlates with the expression of key functional (synapse-related and ion channel) genes and performed high-plex and multiplex in situ hybridization approaches to visualize specific examples of the main populations. Importantly, our taxonomy resisted the test of comparison with prior human striatal snRNA-seq datasets, solidifying its status as a valid consensus classification of striatal interneurons.

## Results

### Interneuron heterogeneity in the human dorsal striatum

With the objective of further decoding the diversity of interneurons in the human dorsal striatum, we isolated and sequenced single nuclei from fresh frozen CN ($N = 25$) and Pu ($N = 28$) samples of 28 control donors who did not meet diagnostic criteria for any neurodegenerative disease (Supplementary Dataset 1). Samples were processed using an established snRNA-seq workflow, which allowed the enrichment of the neuronal population by applying fluorescent-activated nuclei sorting. We also applied spatial transcriptomics (CN and Pu from four donors) and high-sensitivity fluorescent in situ hybridization (FISH, Pu from six donors) in a subset of samples including both sexes (Fig. 1A).

The sequencing yielded 455,886 nuclei, out of which we discarded 29.4% after a thorough quality control process (Supplementary Fig. 1). From the remaining nuclei, we selected the interneurons through an iterative classification process in which we discarded glial cells, MSNs, and excitatory neurons based on bona-fide markers—astrocytes (*AQP4*, *ADGRV1*), microglia (*CSF1R*, *FYB1*), oligodendrocytes (*MBP*, *MOG*, *MAG*), oligodendrocyte precursor cells (*PTPRZ1*, *PDGFRA*, *VCAN*), vascular cells (*EBF1*, *ABCB1*, *ABCA9*), MSNs (*PPP1R1B*, *DRD1*, *DRD2*), and excitatory neurons (*SLC17A7*)—and selected positively for nuclei expressing *GAD1* and/or *GAD2 and/or CHAT*. This classification process resulted in 19,339 nuclei labeled as interneurons, representing the largest dataset of human interneurons from the dorsal striatum available to date (Fig. 1B, C). The interneuron population represented 10.67 % of the total neuronal cells.

We clustered all the interneurons following standard clustering methods (resolution 0.2, see Methods), obtaining 14 clusters which we identified as 14 different interneuron subclasses based on the expression of unique transcriptomic patterns. Merging highly correlated classes (see Methods), we produced a broader classification with eight main classes, which we named after selected marker genes: (i) CCK/VIP (Adenosine Deaminase RNA Specific B2+ [*ADARB2*], *CCK* +, and *VIP* +); (ii) CCK (*ADARB2*+ and *CCK* +); (iii) PVALB (*PVALB* +); (iv) SST/Glutamate Ionotropic Receptor Kainate Type Subunit 3 (GRIK3) (*SST*+ and *GRIK3* +); (v) SST/NPY (*SST*+ and *NPY* +); (vi) PTHLH (*PTHLH*+ and Opsin 3+ [*OPN3*]); (vii) CHAT (*CHAT*+ and *SLC5A7* +); and (viii) TAC3 (*TAC3*+ and Protein Tyrosine Phosphate Receptor Type K+ [*PTPRK*]). The nuclei assigned to different classes and subclasses can be seen separated from each other when projected on the 2-dimensional uniform manifold approximation projection (UMAP, MClnnes Leland et at 2018) of the data (Fig. 1D). The main transcriptomic patterns distinguishing interneuron subclasses are shown in Fig. 1F heatmap, whereas the complete results of a differential expression analysis at class and subclass levels are provided in Supplementary Dataset 2.

In our dataset, PTHLH and TAC3 constitute the largest interneuron classes, accounting for 28% and 28.6% of all detected interneurons, respectively. Both PTHLH and TAC3 classes contained small subclasses, distinguishable from their respective parent classes by the expression of Monooxygenase Ddh Like 1 (*MOXD1*) and Semaphorin 3 A (*SEMA3A*), respectively. The *CCK* + /*ADARB2*+ expressing cells (CCK and CCK/VIP classes) were equally abundant (28.1% of all interneurons) but exhibited greater heterogeneity, with four subclasses (CCK/VIP, CCK/VIP/C-X-C Motif Chemokine Ligand 14 [CXCL14], CCK, and CCK/

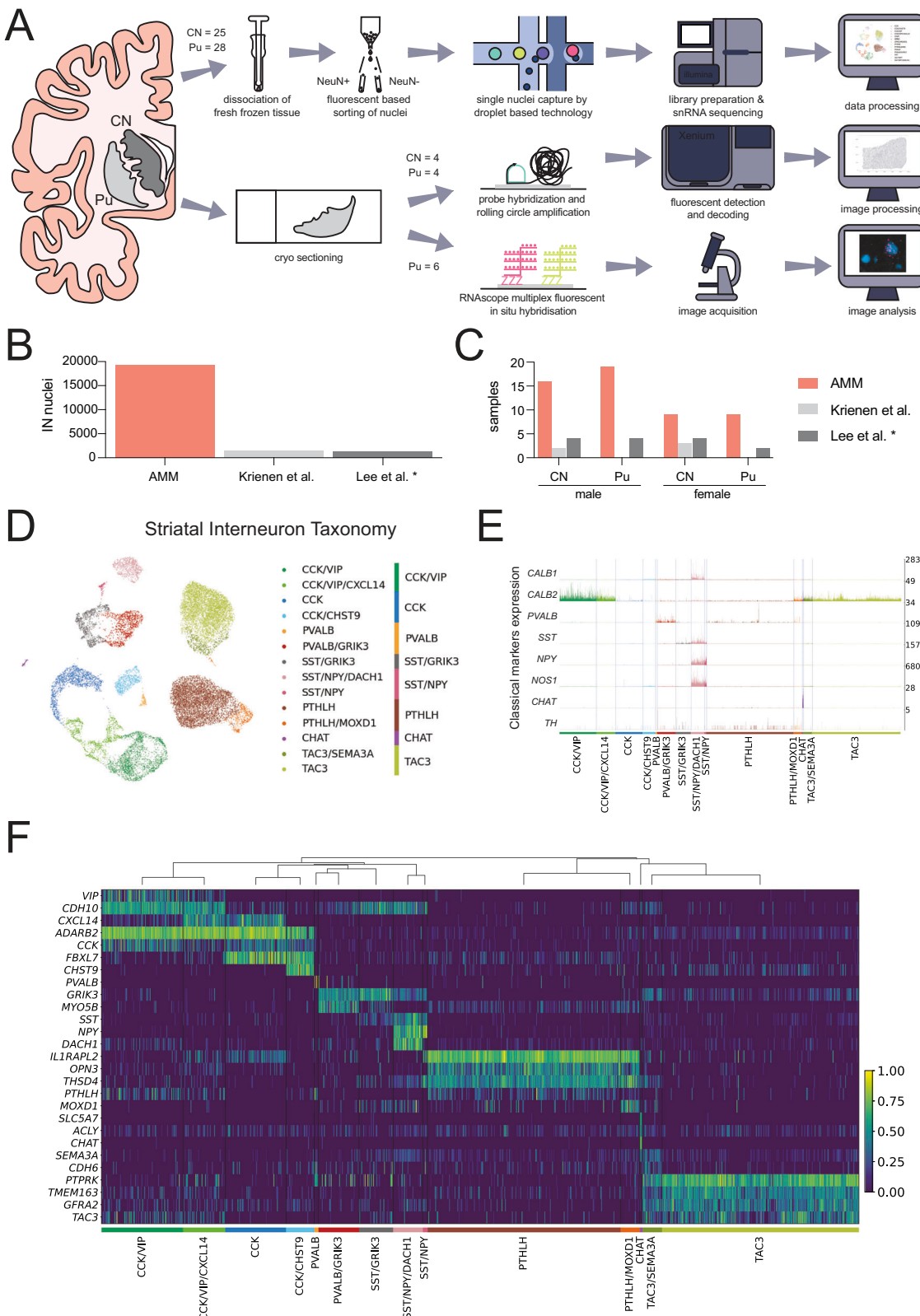

**Nature Communications** | (2024)15:6164

Carbohydrate Sulfotransferase 9 [CHST9]) clearly differentiated by specific marker genes: *VIP*, *CXCL14*, and *CHST9* (Fig. 1F). Although we followed the classical division between CCK and CCK/VIP, we observed that the *ADARB2*+ neurons could also be divided by the expression of the chemokine ligand *CXCL14* and the cadherin 10 (*CDH10*, Supplementary Fig. 2). We also found a great diversity of transcriptomic profiles among the neurons expressing *PVALB* and *SST*, as these two

classes could be divided into five different subclasses based on the expression levels of two specific marker genes: *GRIK3* and Dachshund Family Transcription Factor 1 (*DACH1*). These five subclasses together represent 15% of all interneurons. A more detailed view of classes and subclasses is provided in Supplementary Fig. 3.

Interestingly, we found a smattering of *TAC3* expression in the CCK and CCK/VIP populations, therefore the TAC3 population is best

**Fig. 1 | Interneuron heterogeneity of the human striatum determined by single-nucleus RNA-sequencing. A** Schematic overview of the experimental design. **B** Number of interneuron nuclei sequenced in the present study (labeled as AMM) vs. two other previous works. * Note that the only interneuron nuclei shown from Lee et al.'s study correspond to its control donors. **C** Number of human samples included in this study (AMM) vs. two previous works. * Note that the only samples shown from Lee et al.'s study correspond to its control donors. **D** Uniform manifold approximation projection (UMAP) of the snRNA-seq data of the nuclei labeled as interneurons, colored by interneuron subclass. The corresponding main class is indicated on the right. **E** Normalized expression of classical interneuron marker genes by each of the 14 interneuron subclasses identified. The expression corresponding to each interneuron subclass is colored following the same color scheme as in panel (**D**). **F** Heatmap showing the expression of selected marker genes for each of the 14 interneuron subclasses identified. The expression of each gene is normalized by its maximum value across all nuclei. The dendrogram above the heatmap indicates the proximity across subclasses based on the average Pearson correlation coefficient across all genes between each pair of subclasses. CN caudate nucleus, IN interneuron, Pu putamen.

defined by its high expression level of *PTPRK*. Similarly, we also noted low levels of *PTHLH* in the CCK and the CCK/VIP populations. Therefore, although we decided to keep the PTHLH name to maintain consistency with the nomenclature of mouse interneurons, we identified *OPN3* as a more specific marker gene for the PTHLH cells in the human striatum.

To investigate the alignment of our classification with the traditional taxonomies, we explored the expression of several "classical" markers (*PVALB*, *CALB1* [encoding calbindin], *CALB2* [encoding calretinin], *NOS1* [encoding nNOS], *SST*, *NPY*, *TH*, and *CHAT*), which have been previously used to define the striatal interneuron diversity[15,17] (Fig. 1E and Supplementary Fig. 3). We observed that several of these markers span across various subtypes of the interneurons described here: *CALB2* is highly expressed in the CCK/VIP class and found with lower expression in the TAC3 class, whereas *CALB1* is present in the CCK/CHST9 subclass and had higher level of expression in the SST/NPY/DACH1 subclass. The classical nitrergic cells are represented by both SST/NPY and SST/NPY/DACH1 subclasses as they both express *NOS1* as well as *NPY* and *SST*. Similarly, both the PVALB and PVALB/GRIK3 subclasses are the *PVALB*-expressing cells. The cholinergic cells (here referred to as the CHAT class) were clearly identified, as they present a unique transcriptomic profile. Therefore, our snRNA-seq analysis expands on the classical division since the interneuron types identified by the expression of classical markers can be further divided into classes and subclasses based on their complete transcriptomic profiles. The expression of each of the classical interneuron markers is shown in a UMAP in Supplementary Fig. 3.

Notably, the unheard-of PTHLH population could not be characterized by any of the classical markers, although it exhibited low levels of *PVALB* (as recently demonstrated in the mouse striatum[28]) and very low and sparse *TH* expression (Fig. 1E). Similarly, we did not detect any classical markers unique to the TAC3 cells, which were recently identified as primate-specific[37].

Regarding the classical TH interneurons, we did not find an interneuron class or subclass characterized by *TH* expression. This is in agreement with the findings of previous snRNA-seq studies, where *TH* expression in interneurons was also negligible[37–39]. Moreover, the low abundance of TH+ interneurons (<1%) found with immunohistochemical techniques is also in line with these observations[33].

Finally, although our SST/GRIK3 interneurons expressed the classical marker *SST*, they did not co-express the marker genes *NOS1* and *NPY* classically associated with *SST*+ interneurons. Instead, *SST*+ interneurons in this cell class were characterized by the expression of the glutamate receptor *GRIK3* as well as *TAC1* and *TAC3* (Fig. 1F).

We next asked whether there could be an age-related shift in proportions of the various interneuron classes and subclasses. Since we lacked the statistical power to establish a correlation between the abundance of each interneuron population and age as continuous variable, we binned donors' age in <50, 50–70, 70–90 and >90 years old. We observed that all interneuron classes and subclasses were present in samples from all the age groups in our dataset (Supplementary Fig. 4). Thus, we clearly determined that there was no class or subclass exclusive of any age group.

## Interneuron populations exhibit region-based differences within the striatum

CN and Pu have different inputs and projections within the basal ganglia circuit[40,41]. To examine possible regional differences in CN vs. Pu interneurons, we next compared both the proportions and transcriptomic profile of interneurons from these two striatal regions at the class level. We chose to focus on classes rather than subclasses to increase the robustness of our results, as the low numbers of some of the subclasses would limit the reliability of these analyzes. While we found all the interneuron classes identified in our snRNA-seq data in both CN and Pu (Fig. 2A), we did note slight differences in abundance between both regions: CN was significantly richer in PTHLH interneurons (35.6% vs. 20.3%, *p*-value = 0.001), whereas the CCK, SST/GRIK3, and PVALB classes were significantly more abundant in the Pu (12.1% vs. 7.1%, 3.5% vs. 2.4%, and 5.5% vs. 2.7%, respectively, with *p*-values 0.008, 0.027, and 0.015, respectively; Fig. 2C). Several interneuron classes also exhibited distinct region-dependent transcriptomic signatures. Most notably, we found that the PTHLH class had significant differences in the expression of 276 genes by region (i.e., 126 upregulated and 150 downregulated in CN vs. Pu). SST/NPY, CHAT, TAC3, and CCK/VIP classes also showed expression differences in CN vs. Pu, but of smaller magnitude (Fig. 2B, Supplementary Dataset 3).

To contextualize the changes in expression of the PTHLH class neurons across the two striatal regions, we conducted a gene-set enrichment analysis using the genes significantly upregulated in either region (Fig. 2D). The 126 genes upregulated in the CN vs. Pu were enriched in Gene Ontology (GO)[42]-terms associated with secretory vesicles and their transport, whereas the 150 genes with significantly higher expression in the Pu vs. CN were enriched in GO-terms associated with receptor complexes, ion channel complexes, plasma membrane components, and G-protein-coupled receptor (GPCR) activity. This signal transduction via GPCRs relies upon the production of cyclic Adenosine Monophosphate (cAMP) and other signaling cascades[43]. Of note, the Kyoto Encyclopedia of Genes and Genomes (KEGG) pathways showed that the differentially upregulated genes in Pu vs. CN are related to the cAMP signaling pathway, including genes such as *ADCY8* (a GPCR), *CRHR1*, *GRIA3*, *GRIN3A*, *PDE4B*, *PLCE1*, and *RYR2*. Thus, these data suggest that there is an over-expression of alpha-amino-3-Hydroxy-5-Methyl-4-Isoxazolepropionic Acid (AMPA) and N-Methyl-D-Aspartate (NMDA) receptor subunits (related to $Ca^{2+}$ and $Na^+$ flux) in Pu compared to CN via cAMP/GPCRs activation, which could lead to enhanced long-term potentiation and higher synaptic plasticity in Pu vs. CN and explain why CN and Pu inputs and functionalities are not analogous[44].

Hence, our snRNA-seq approach revealed subtle yet significant variations in the proportion of key interneuron classes with potentially relevant implications for the functioning of the basal ganglia circuit.

## Spatial transcriptomics profiling and FISH validate the interneuron subclasses defined via snRNA-seq

Transcriptomic data obtained via snRNA-seq lack spatial information, which is valuable to understand the physical relationships between

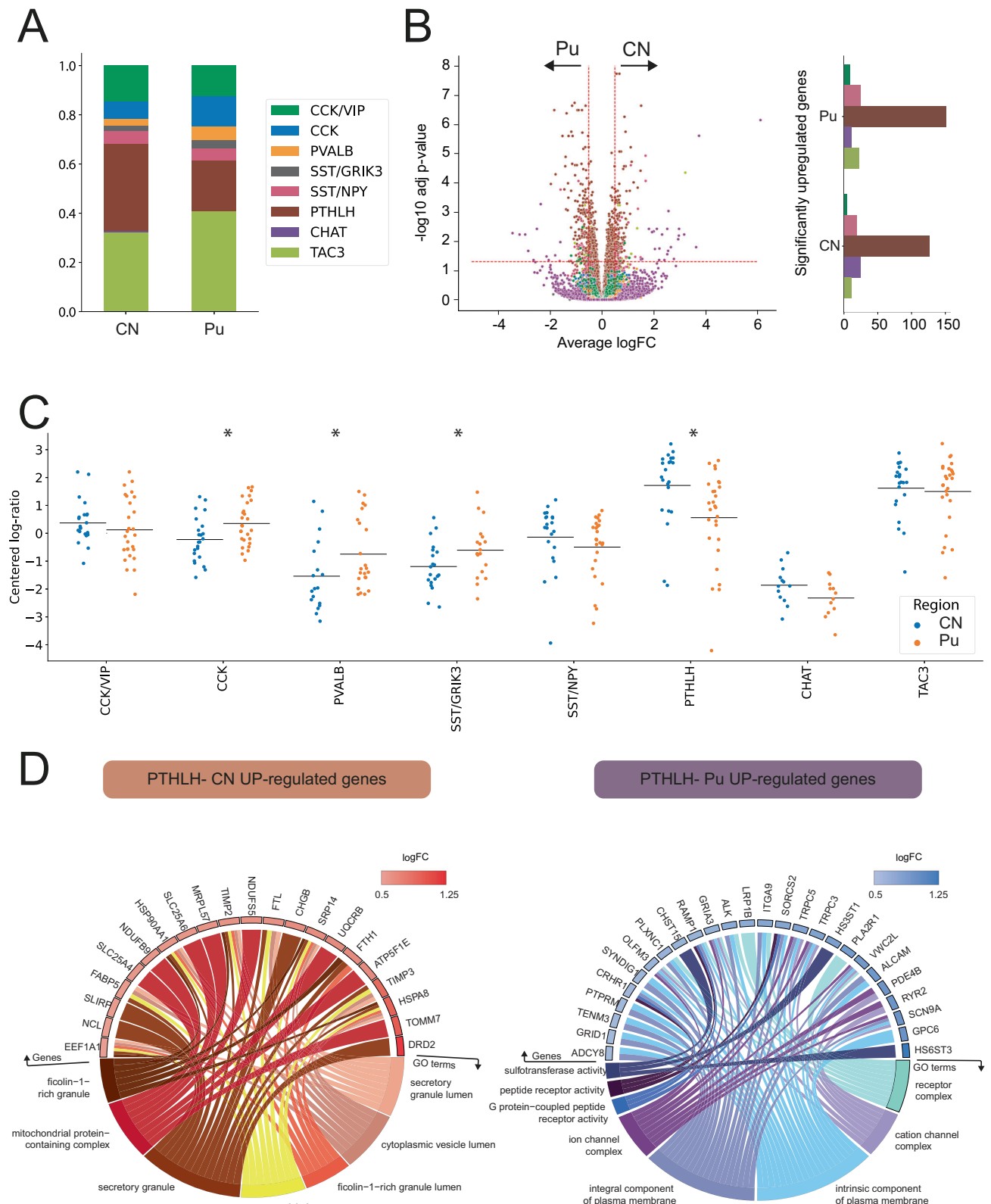

different cell types and/or different classes within a cell type. To confirm the existence of this diversity of interneuron populations in the spatial context of the human striatum, we used the Xenium in situ transcriptomic platform, which enabled us to comprehensively characterize the expression profile of 366 target genes in tissue sections from both CN and Pu with subcellular resolution. Based on their main marker genes, we were able to validate in situ the 14 subclasses

described by snRNA-seq among 8,096 identified interneurons. The combination of marker genes used to first identify all neuronal cells and then annotate the interneurons into specific classes are detailed in the Methods section. To study the abundance of the different classes and subclasses we calculated the relative proportions of each of them (Fig. 3B,C). Consistent with what we observed in the snRNA-seq dataset (Figs. 1F and 2A), our spatial transcriptomic analysis revealed that

**Fig. 2 | Striatal interneuron classes exhibit region-specific differences. A** Barplot illustrating the different proportions of interneuron subclasses in CN and Pu. **B** (left) Volcano plot showing differentially expressed genes (DEGs) for each subclass. DEGs with an adjusted *p*-value < 0.05 (likelihood ratio test, Benjamini-Hochberg correction) and an average logFC > 0.5 were selected. (right) Barplot indicating the number of significantly upregulated genes per interneuron class on each region with bars colored following the same color scheme as the legend in panel (**A**). **C** Scatter dot plot representing the compositional analysis of inter-neurons in CN vs Pu (estimated by centered log-ratio method) and demonstrating

significant compositional differences between CN and Pu in CCK (*p* = 0.008), PVALB (*p* = 0.015), SST/GRIK3 (*p* = 0.027), and PTHLH (*p* = 0.001) interneurons (two-sided Wilcoxon test). **D** Circos plots showing the GO-term enrichment analysis of upregulated genes in PTHLH subpopulation in CN (left) and Pu (right). Each GO circos plot illustrates enriched terms (adjusted *p*-value < 0.05, hypergeometric test, Benjamini–Hochberg correction) with their respective enriched genes along with the logFC of these genes. * *p* < 0.05. CN caudate nucleus, Pu putamen.

PTHLH and TAC3 subclasses are the most abundant in both CN and Pu, constituting 21.2 and 20.9% of the total interneurons respectively (Fig. 3B). In contrast, the PTHLH/MOXD1 and TAC3/SEMA3A subclasses were less abundant, accounting for 6.2% and 12.5% in CN and 9.0 and 11.0% in Pu, respectively. Conversely, the CCK subclasses, including CCK, CCK/CHST9, CCK/VIP, and CCK/VIP/CXCL14, along with the previously undescribed SST/GRIK3 subclass, represented the rarest interneuron populations, with frequencies ranging from 0.68% to 1.46%. Similarly, the PVALB and PVALB/GRIK3 subclasses exhibited comparable presence in both CN and Pu (1.29% and 1.8% for the former and 2.2% and 3.7% for the latter). Nitrergic interneurons, represented by SST/NPY and SST/NPY/DACH1 subclasses, showed an intermediate proportion between the aforementioned extremes, accounting for 7.5% and 7.6%, respectively. Interestingly, we detected significantly more CHAT cells in our spatial transcriptomic study than in our snRNA-seq data (13.3% vs. 0.3%, respectively). This discrepancy might indicate a technique-specific bias for this specific subtype.

Once cell types or states were annotated in the spatial transcriptomic dataset, we quantified the spatial enrichment of the different subclasses and explored cellular neighborhoods throughout the tissue (Fig. 3D). Specifically, we calculated a spatial proximity enrichment score based on the frequency of observations from one cluster being close to observations from another, with a high score indicating enriched relationships. Although the overall scores were not very high, we noticed slight spatial preferences. For example, SST/NPY and SST/NPY/DACH1 interneurons were located closer to PVALB/GRIK3, while CHAT interneurons were closer to the PTHLH/MOXD1 subclass and to the CCK and PVALB classes.

In addition to the spatial transcriptomics approach, we sought to further validate some of the described subclasses through quantitative multiplex FISH using up to three marker genes in the Pu of six of the snRNA-seq study donors. Using different combination of probes against the main marker genes (*PVALB, PTHLH, OPN3, CCK, ADARB2, VIP, TAC3,* Thyrotropin Releasing Hormone [*TRH*], *SST, NPY* and *DACH1*), we identified examples of cells belonging to different classes or subclasses, including PVALB/GRIK3 (*PVALB*+ and *GRIK3* + ), PTHLH (labeled with *PTHLH* and *PVALB* or *PTHLH* and *OPN3*), both CCK classes (co-expressing *CCK, ADARB2,* and *CDH9*), CCK/VIP (*CCK*+ and *VIP* + ), TAC3 (*TAC3*+ and *TRH* + ), and SST/NPY/DACH1 (using *SST, NPY,* and *DACH1* probes) (Fig. 3F and Supplementary Fig. 5). For three of the classes (PTHLH, PVALB, and SST/NPY) we performed a deeper analysis including quantifications (Supplementary Fig. 5). Notably, the *SST/NPY/DACH1* FISH revealed cells double-positive for *SST* and *NPY* and cells triple-positive for *SST, NPY,* and *DACH1* (Supplementary Fig. 5B–D), which agrees with both our snRNA-seq data (Supplementary Fig. 5A) and our spatial transcriptomics results (Fig. 3E). In addition, we identified a group of cells that were only positive for *SST* and most likely correspond to the SST/GRIK3 subclass. The *PVALB*-expressing interneuron subclasses are of particular interest because *PVALB* has traditionally been used to identify a class of striatal inter-neurons. In contrast to the mouse striatum, in which *Pvalb*+ inter-neurons were reported to be contained within the *Pthlh*+ cells[20] (i.e., all *Pvalb*+ interneurons are *Pthlh* + ), our snRNA-seq data from human CN and Pu indicates the presence of a distinct *PVALB*-positive but *PTHLH*-negative subclass of interneurons. FISH using *PTHLH* and *PVALB*

probes revealed *PTHLH* single-positive cells, or with low *PVALB* expression, in all six donors analyzed, while *PVALB* single-positive cells were found in five of the six donors (Supplementary Fig. 5E, F). These results are again in agreement with our spatial transcriptomics and snRNA-seq data, where the PVALB class shows high *PVALB* expression levels but it is one of the least abundant subclasses, whereas the PTHLH class is much more abundant but expresses low levels of *PVALB* (Figs. 1E, F and 2B, C).

In summary, overall, we were able to confirm our snRNA-seq-based classification of striatal interneurons with two other orthogonal methods of spatial gene expression analysis, thus reinforcing the validity of this taxonomy.

## PTHLH and TAC3 subclasses exhibit changes along continuous transcriptomic profiles

Our initial cluster analysis allowed us to detect 14 different interneuron subclasses, each characterized by the expression of a unique combination of marker genes. However, previous studies have shown that striatal interneurons display gene expression gradients[28–30] and, therefore, their diversity may not be captured by discrete classifications. To investigate if this phenomenon was observable in our dataset, we conducted a factor analysis within the largest subclasses (PTHLH and TAC3) on each striatal region. In both subclasses and regions, the factor analysis revealed coordinated gradual changes of sets of genes (Fig. 4A–D, left and middle panels). The genes with the highest weights on the factor describing the differences within each population were different across regions (Fig. 4A–D, right panel, Supplementary Dataset 4), although there were some commonalities; for example, large changes in *SLIT1, CNTN5,* and *TAFA2* expression were observed in the PTHLH neurons in both Pu and CN. Similarly, *KCNIP4, ASIC2,* and *MARCH1* were responsible for some of the largest variations across the TAC3 neurons in both striatal regions. Notably, some of the genes with the greatest contributions to the intra-subclass variance (*KCNIP4, ASIC2, RYR2*) are ion channel subunits, suggesting the possible existence of different electrophysiological phenotypes within the same transcriptomic subclass. To gain a better understanding of the differences revealed by the factor analysis, we conducted a gene set enrichment analysis on the genes with the largest contributions to the factor on each case (Supplementary Figs. 6 and 7).

The genes of the TAC3 class associated with the gradient driven by *KCNIP4* and *ASIC2* expression levels showed similar enriched terms in both CN and Pu (Supplementary Fig. 4). Relevant terms were mostly related to the regulation of synapse formation or activity and cell adhesion. Genes of the PTHLH class related to the gradient driven by *GULP1, ZNF385D,* and *RYR2* expression levels in the CN (*CDH8, PDE4B,* and *FRAS1* in Pu) displayed terms linked to cell adhesion and channel complexes, specifically $Ca^{2+}$ channels (Supplementary Fig. 6). However, results for genes from the Pu gradient defined by *CNTN5, TAFA2,* and *SLIT1* showed biological specificity: GO-term enrichment analysis uncovered functionalities in synapse organization and ion transporter activity via ion channels that appear to be specific to the Pu.

These analyzes suggest that PTHLH and TAC3 interneurons are not transcriptionally uniform but exhibit gradients of expression of synaptic and ion-channel genes, with potentially relevant functional implications.

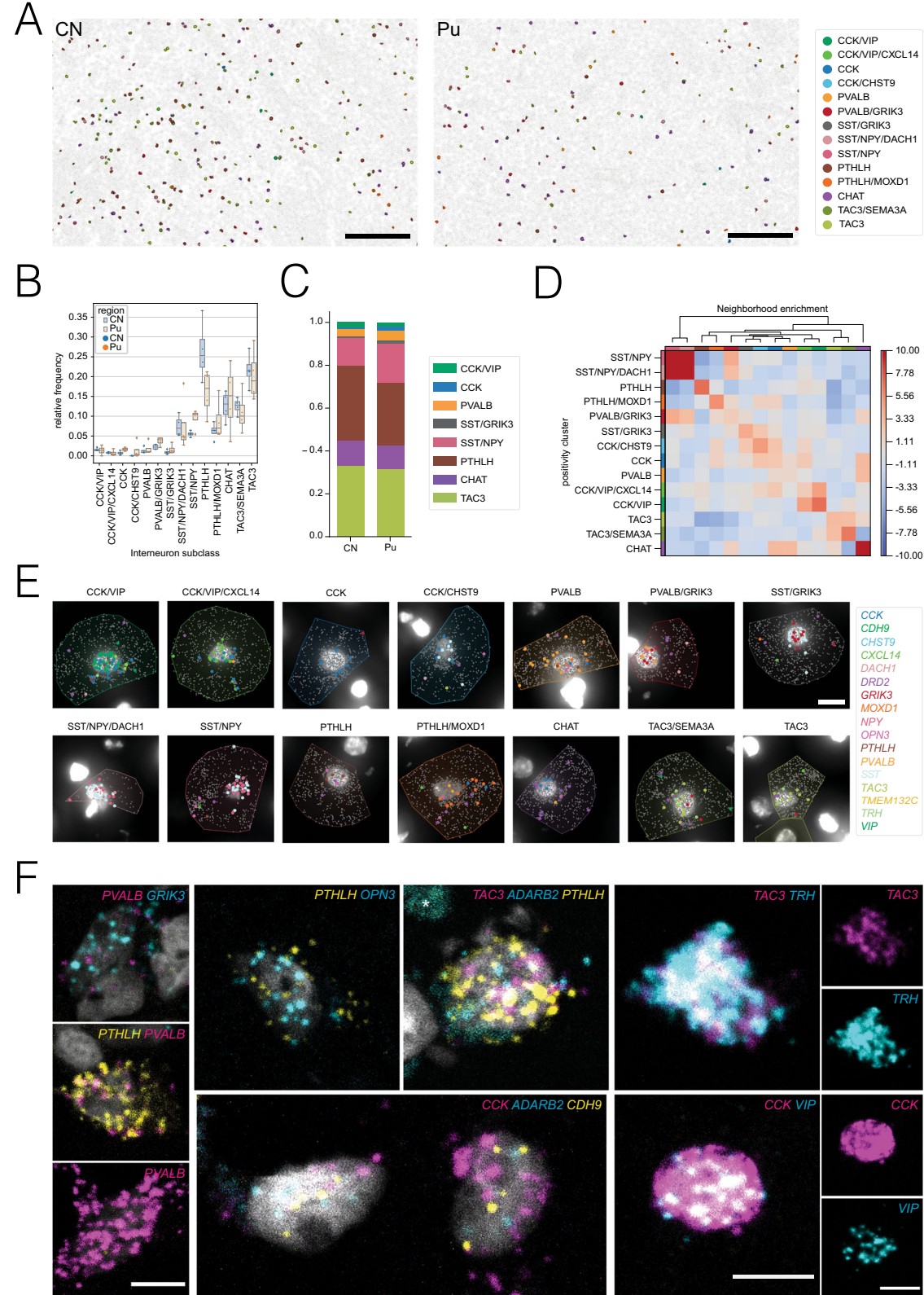

## Interneuron taxonomy is maintained across functionally relevant genes

Besides the expression of specific markers, human striatal interneurons have been traditionally classified based on their electrophysiological properties. To further understand the potential functional implications of our taxonomy, we investigated the differences existing between the established subclasses across two separate sets of genes highly relevant to neuronal function. First, we restricted our dataset to the genes corresponding to dopamine, GABA, acetylcholine, and glutamate receptors. The UMAP of our data on this set of genes shows a clear separation between the different subclasses (Fig. 5A, left) and the differential expression analysis revealed unique neurotransmitter-receptor expression patterns (Supplementary Fig. 8, Supplementary Dataset 5). Noteworthy markers fitting this pattern

**Fig. 3 | Xenium-based in situ transcriptomics profiling and FISH validation.**
**A** Overview for CN and Pu sections after segmentation and cell type classification, scale bar 200 µm. Segmented cells are represented as polygons, based on their expanded segmentation masks. For interneurons, polygons are colored based on their assigned subclass, while non-interneuronal cells are outlined in gray.
**B** Stripplot depicting the relative frequency of interneuron subclasses with variance per sample in CN and Pu ($N = 4$ each). On top, boxplots are overlaid with minimum and maximum values represented by the whiskers' ends, while percentiles 25, 50, and 75 are defined by the position of the box. **C** Barplot illustrating the different proportions of interneuron main classes in CN and Pu ($N = 4$ each). **D** Heatmap representing the spatial neighborhood enrichment between interneuron populations characterized across the different sections ($N = 8$). **E** Representative images for 14 interneuron subclasses, identified in the Xenium experiment, scale bar 10 µm. DAPI staining is shown as a grayscale image. Read detected on each cell are represented as individual dots, colored based on their transcript identity. Expended segmentation masks of the outlined cells are colored based on their assigned interneuron subclass following the colormap specified in Fig. 3A. **F** Images of various in-situ hybridization experiments illustrating 10 proposed marker genes, asterisks mark autofluorescence due to lipofuscin, scale bars 10 µm.

were: *GRIN3A* and *GRM5* in CCK interneuron class; *GRM7*, *CHRM3*, *GABRA1*, and *CHRNA2* in CCK/VIP; *GRIN2C* in PVALB; *GRIK3*, *GRIK1*, and *GRM1* in SST/GRIK3; *GRIP1* and *GRIA4* in PTHLH; *CHRM2* and *DRD2* in CHAT; and *GRM8, GRID1, CHRM2*, and *CHRNA7* in TAC3 interneurons.

We conducted a similar analysis using all the genes under the GO-term "ion channel activity" (GO:0005216). The UMAP of the data using only ion channels retained the separation between subclasses (Fig. 5A, right), whereas the differential expression analysis rendered unique transcriptomic patterns. Relevant marker genes related to these patterns were: *KCNIP1, CACNA2D1*, and *KCNH5* in CCK; *GLRA2* and *KCNT2* in CCK/VIP; *KCNMB4* and *RYR1* in PVALB; *GRID2* in SST/GRIK3; *KCNMA1, GLRA3*, and *ITPR2* in PTHLH; *TRPC3* and *KCNG3* in CHAT; and *GRID1, SCN7A*, and *CACNA2D3* in TAC3 interneurons (Supplementary Fig. 8, Supplementary Dataset 5).

A closer inspection of this functionally relevant analysis revealed a strong parallelism in key feature genes between the TAC3 population, which has been recently described as a primate-specific class[37] and we have thoroughly characterized here, and the mouse Th interneuron class[15,28]. Indeed, we found notable gene expression similarities when comparing our human TAC3 class with the mouse Th class from dataset A in ref. 28. Both classes share the expression of the Tachykinin precursor—*TAC3* in human and the homologous gene *Tac2* in mouse—and the *TRH* —recently described as a marker for the mouse Th population[28]. Additionally, they share the cholinergic nicotinic receptor subunits *CHRNA3/Chrna3* and *CHRNB4/Chrnb4*, the Glial Cell Derived Neurotrophic Factor (GDNF) receptor *GFRA2/Gfra2*, the prolactin receptor (*PRLR/Prlr*) and the serotonergic receptor HTR3A/*Htr3a*. Interestingly, they also match in their negative expression patterns, such as the absence of expression of both Synaptotagmin 1 (*SYT1/Syt1*) and the glutamatergic receptor subunit *GRIK1/Grik1*, which are remarkably highly expressed in the rest of interneuron populations in both human and mouse (Fig. 5B). Of note, as shown in Fig. 1E, we could hardly detect *TH* expression in interneurons, but we found it in MSNs, mainly in those expressing Dopamine Receptor D1 (*DRD1*) (Supplementary Fig. 9A). To validate this observation, we performed FISH on Pu samples from six donors, combining *TH* and *DRD1* probes. We observed that among *TH*+ cells, 81.9% also express *DRD1* (Supplementary Fig. 9B, C), consistent with our snRNA-seq data. Moreover, we investigated *TH* expression in other publicly available striatal snRNA-seq datasets and confirmed that, whenever detected, *TH* expression predominantly maps with MSNs rather than interneuron markers (Supplementary Fig. 9D).

Lastly, to determine what drives the expression of ion channels in these interneuron subclasses, we focused on the typical genes of a Fast Spiking (FS) profile characteristic of high *PVALB*-expressing cells (Fig. 5C). This analysis showed significant expression of FS genes[45–48] such as *KCNAB1* (Kvb1.3), *KCNC2* (Kv3.2), *KCNA2* (Kv1.2), *KCNC1* (Kv3.1), *HCN1*, and *SCN1A* (Nav1.1) in PTHLH and PVALB cells, with a substantial overexpression in the latter (Fig. 5C). In the mouse Pthlh population, the FS profile correlated positively with the *Pvalb* expression level following a continuous gradient pattern[28]. Interestingly, in the human striatum, we also found a defined PVALB class of interneurons with high expression of the genes involved in the FS profile.

Taken together, these results show a remarkable correlation between the gene expression profiles and previously reported electrophysiological features of striatal interneuron classes, supporting the idea that distinct transcriptional signatures contribute to the long-known electrophysiological diversity of striatal interneurons.

## Interneuron taxonomy is consistent across published human striatal snRNA-seq datasets

To further validate our findings and test the generalizability of our classification of interneuron subclasses in the human striatum, we integrated our labeled data with four other datasets from three sources[37–39]. These datasets included CN[37,39], Pu[39], and nucleus accumbens[38] human samples (Supplementary Dataset 6). We filtered and normalized the raw counts from these four datasets and selected the interneuron nuclei based on the same markers as in our own dataset (see Methods). This resulted in a total of 8,090 additional interneuron nuclei added to our 19,339. We reduced the ensemble of the five datasets to the 12,986 overlapping genes, of which we selected the top 1,200 most variable to build an integrated model using Single-Cell Variational Inference (scVI)[49].

Remarkably, the UMAP of the integrated data revealed extensive overlap between nuclei from different datasets, indicating that the scVI model compensated possible batch-specific differences (Fig. 6A). Clustering the integrated data resulted in 16 groups (Fig. 6B), out of which 15 had a clear correspondence to our original 14 interneuron subclass labels (Fig. 6C). Only cluster number 12 contained less than 1% of nuclei from our dataset and could not be readily matched to any of the described subclasses. This cluster consisted of 550 nuclei (2% of the total), 83.6% of which belonged to the DropSeq dataset from Krienen et al.'s study[37]. To ensure that cells clustering together actually shared the same transcriptomic profile, we examined the expression of the subclass marker genes on each of the public datasets (Fig. 6D, Supplementary Fig. 10). As shown in Fig. 6, cells within the same cluster have similar expression patterns, which in turn correspond to one of the subclasses established in our taxonomy. In the case of cluster number 12, we observed that it did express the markers corresponding to our TAC3 subclass. Thus, we could identify each of the clusters on the integrated data as one of the 14 interneuron subtypes in our proposed taxonomy. Although subclass proportions differed by dataset, all subclasses contained nuclei from at least two different datasets (Fig. 6E).

In summary, cross-validation with prior smaller snRNA-seq studies in human striatum supports the generalizability of our taxonomy.

## Discussion

The neuronal communication in the striatum, a hub for motor and cognitive information, is modulated by the interneurons. Characterizing the diversity and abundance of these locally-projecting neurons is key to understand the proper functionality of this brain structure. Here we produced the largest snRNA-seq dataset of the human dorsal striatum (both in number of nuclei isolated and human samples) to date to profile the interneuron diversity of this brain structure (CN and Pu) together with the highly sensitive and recently developed spatial

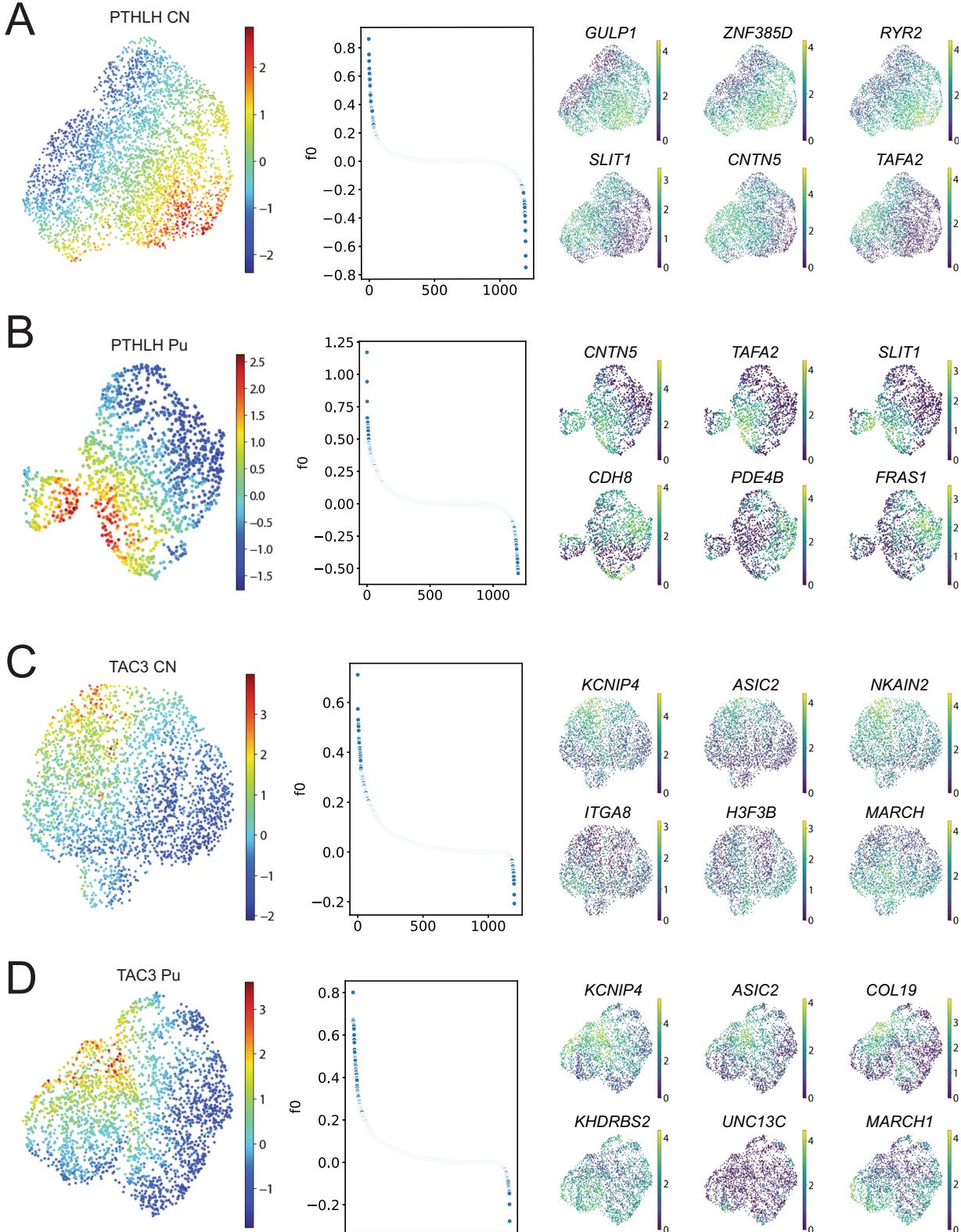

**Fig. 4 | Factor analysis within interneuron subclasses. A** (left) UMAP of the PTHLH subclass from the CN, colored by the value of the factor obtained by running factor analysis. (middle) Factor weights associated with each gene. (right) UMAP of the PTHLH subclass interneurons colored by the expression level of (top row) the genes with the top three and (bottom row) bottom three weights on the factor. **B–D** (left) Factor values, (middle) weights distributions and (right) expression of genes with largest contributions to the factor obtained for the PTHLH subclass from the Pu, the TAC3 subclass from the CN, and the TAC3 subclass from the Pu, respectively. CN caudate nucleus, Pu putamen.

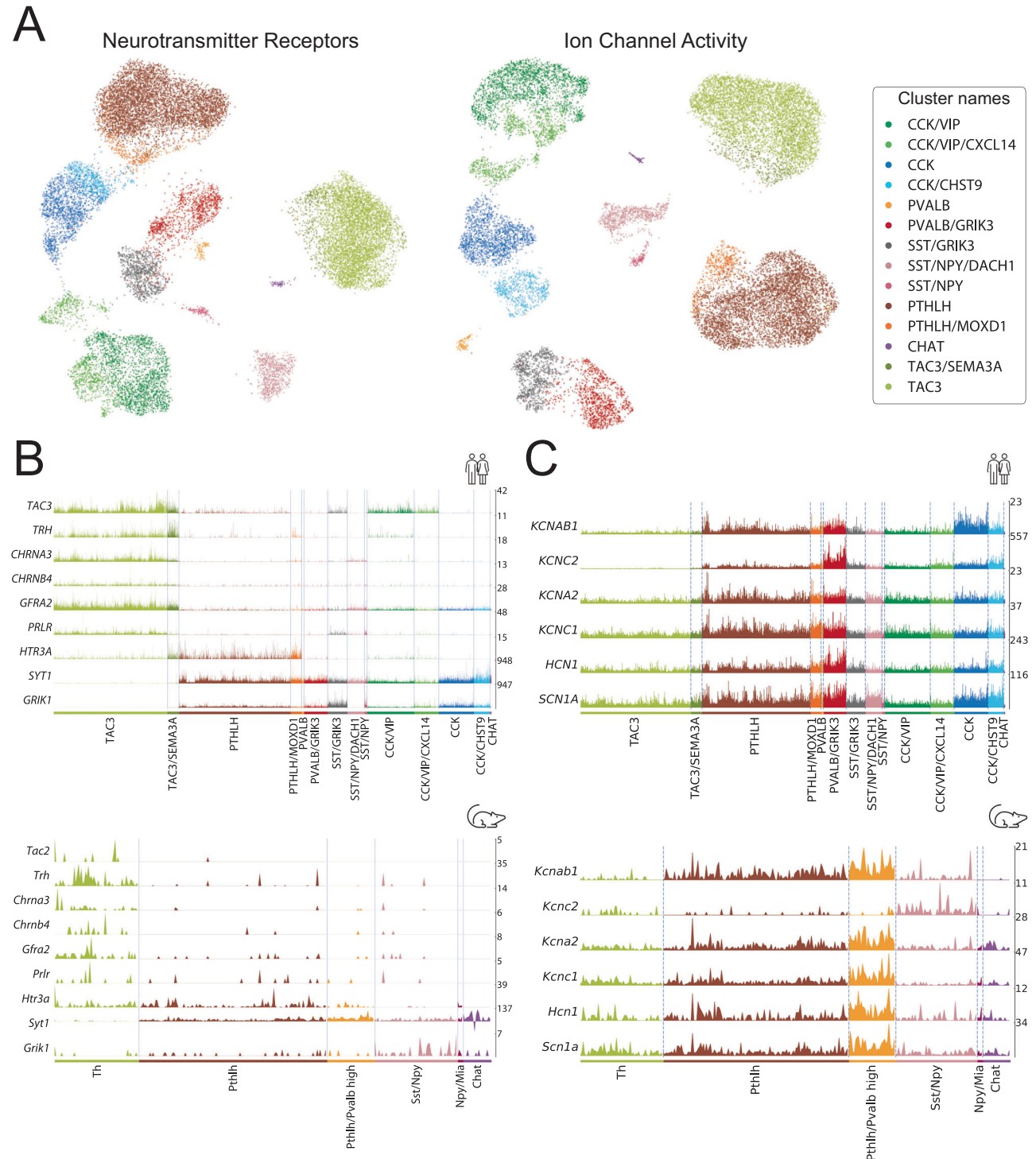

**Fig. 5 | Comparison of striatal interneuron subclasses between mouse and human. A** (left) UMAP of the interneuron nuclei using expression data restricted to neurotransmitter receptor genes. (right) UMAP of the same data restricted to the genes annotated with the molecular function "ion channel activity" (GO:0005216). Colored based on the interneuron classification established before.

**B** (top) Expression of raw values of genes suggesting parallelisms between the TAC3 subclass in the present human dataset and (bottom) the Th interneurons in the mouse striatum described by Muñoz-Manchado et al. [28]. **C** (top) Expression values of genes related to the fast-spiking phenotype in this human striatal dataset vs. (bottom) the striatal mouse dataset from Muñoz-Manchado et al. [28].

transcriptomics approach Xenium (10XGenomics)[50–52]. We leveraged this dataset to perform a deep molecular characterization of the 14 interneuron subclasses identified, provide a full set of marker genes for each one, and delineate functional aspects such as synapses-related machinery for different classes, differences between CN and Pu, inner gradient structure of gene expression levels, and relevant pathways

related to the main gene expression differences. Importantly, these results pertain to a specific region of CN and Pu (see Methods). Slight differences along the anterior-posterior and dorso-ventral axes are expected, as previously described.

The interneuron diversity of the mammalian striatum has received little attention until recently, especially if we compare it with other

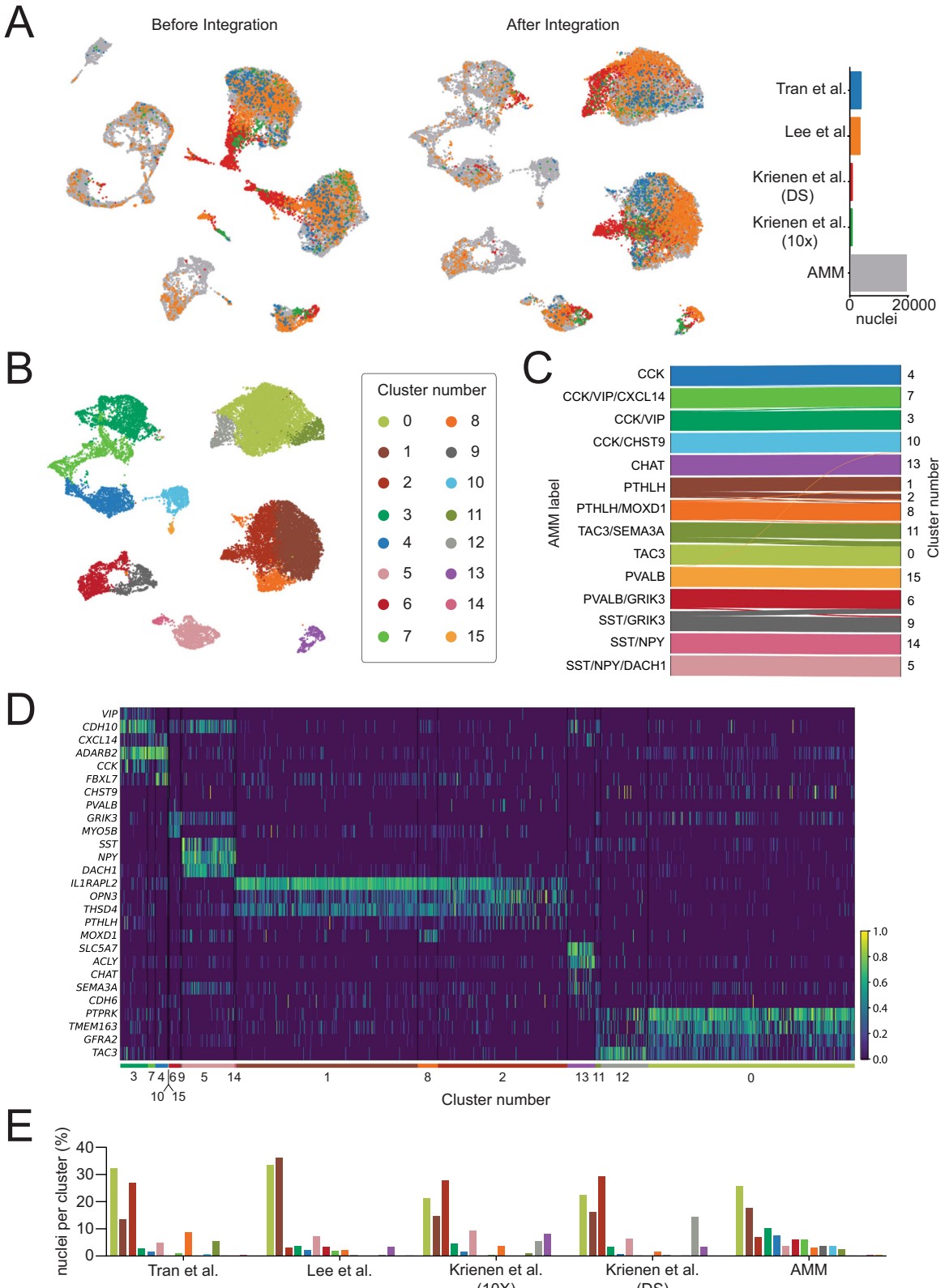

**Fig. 6 | Interneuron taxonomy is consistent across multiple human striatal snRNA-seq datasets. A** (left) UMAP of interneuron nuclei from five different datasets before and (middle) after integration with scVI. (right) Barplot indicating the total number of nuclei from each dataset. **B** UMAP of the integrated data colored by cluster. **C** Shankey diagram relating the labels of the nuclei in the AMM dataset to the clusters obtained on the integrated data. Only assignments with more than 1% of the cells of each subclass are shown. **D** Normalized expression of interneuron subclass marker genes on the integrated public datasets (excluding our own). **E** On each dataset (x-axis), each bar represents the percentage of all interneuron nuclei detected in that dataset (y-axis) that belong to a specific cluster color-coded as in (**C**).

brain regions, such as the cortex, where numerous investigations have been carried out[53]. Classically, striatal interneurons have been identified according to six main markers or combination of markers: CR, CALB1, TH, PVALB, CHAT, and nNOS/NPY/SST[15–17].

With the development of snRNA-seq, many studies have contributed to elucidate the cellular composition of different brain areas, including the striatum, especially in the mouse brain. This technology offers a full genetic delineation to characterize molecular cell identities in a systematic manner. Only a few published snRNA-seq datasets (with no high-plex spatial transcriptomic approach) contain information regarding the human striatum, but they captured a low number of interneurons as depicted in Fig. 1B and/or focused on other cell types, thus precluding the establishment of a comprehensive taxonomy, as suggested by their lack of agreement[37–39] Krienen et al.[37] identified seven types of striatal cells, with six found in humans, based on specific markers: *SST+*, *CHAT+*, *ADARB2+*, *PTHLH+*, *TAC3+*, and *TH+* (although no *TH* expression was detected), Tran et al.[38] focused on the nucleus accumbens and identified five interneuron types (A to E) with distinctive marker expressions, *SST*, *NPY*, *VIP*, *TAC3*, and two classes suggested as *PV*-expressing interneurons (despite no robust *PV* expression observed), but did not detect cholinergic or TH interneurons. Lee et al.[39] provided a snRNA-seq dataset from the human striatum, distinguishing three interneuron classes (PVALB/TH, SST/NPY, and CHAT) without further detailed exploration.

In this work, we have sampled nearly half a million CN and Pu nuclei, obtaining almost 20,000 high-quality interneuron nuclei after a strict quality control and classification process, a number that enabled us to establish a robust taxonomy of human striatal interneurons.

We found eight main classes that we named after one or two of their main molecular markers. Two of these eight main classes express *CCK* and represent almost one third of the total interneuron population. These two *CCK*+ populations were split into two subclasses based on their additional expression of *VIP*: CCK/VIP and CCK/VIP/CXCL14 for those expressing *VIP* and *CCK*, and CCK and CCK/CHST9 for those which do not, respectively. Of note, they all share the expression of *ADARB2*, which separates them from the rest of interneurons and is a marker used to designate the developmental origin of cortical interneurons from the caudal ganglionic eminence (CGE)[54]. Pertinent investigations would be needed to confirm whether this is also the case in striatum since cortex and striatum seem to present differences in the combinatorial markers of developmental origin[55,56]. CCK and CCK/VIP populations were first reported as striatal interneuron cell classes in a scRNA-seq study of the mouse striatum enriched for interneurons[28]. *Adarb2/Cck-* and *Adarb2/Vip*-expressing interneurons have also been described in the striatum of both marmoset and mouse in a cross-species study[37]. Remarkably, we find a significant increase of CCK-expressing cells in human vs mouse dorsal striatum, which might indicate their greater involvement in highly complex computational processes for motor and cognitive functions in the human. Little is known about the role of *CCK*-expressing cells in the central nervous system[57], although *CCK* is widely used as an interneuron marker in cortical areas. Further investigations would be needed to decipher the role of one of the most abundant interneuron classes in the human striatum.

How does this transcriptomic-based taxonomy of striatal interneurons relate to prior morphology-based taxonomy? The correlation between our classification system and classical markers of interneurons is depicted in Fig. 1E. Despite the power of snRNA-seq to identify transcriptomically distinct subclasses, a significant overlap is evident when comparing our main classes with those from previous studies that integrated immuno-/histochemical and morphological features to define striatal interneuron diversity. For example, Cichetti et al.[17] defined five main classes using protein/enzyme expression (CR, PVALB, CB, NADPH-d, and CHAT), further subdivided based on perikaryal shape and dendritic characteristics. Within the CR class, four

subgroups align with our taxonomy expressing *CALB2* (CR): CCK/VIP, PTHLH (mainly the *MOXD1*+ cells), CHAT, and TAC3. Notably, recent findings indicate that the primate-specific large aspiny CR cells, as described by Cicchetti et al., also express *CHAT*, supporting our results. Examining the human striatal transcriptome, CR seems to label genetically unrelated populations (see tree dendrogram in Fig. 1F), which could potentially explain the varied morphologies observed with immunohistochemistry for CR (CALB2). Regarding PVALB cells, three subdivisions from Cichetti et al. align with our PVALB, PVALB/GRIK3, and low-expression *PVALB*+ cells within the PTHLH population. In contrast, Cichetti et al.'s CALB1 and NADPH-d (or NOS1) cells show three distinct groups, which specifically mapped to the SST/NPY class in our taxonomy, possibly due to regional differences and/or scarcity of one of the morphological types. Lastly, their CHAT cells perfectly match as a single group with our CHAT class. The proportion of CHAT cells we found in our snRNA-seq data was lower than anticipated based on other studies (0.3% versus 11%[33,35]). However, our spatial transcriptomic analysis results closely matched the previous data (13.3%), suggesting a technique-specific bias in the snRNA-seq data. Further studies are required for a more conclusive matching of features, including morphology or other characteristics.

The hierarchical clustering of subclasses in our taxonomy places the *CCK*-expressing cells in the same main branch as a group of cells expressing the glutamatergic receptor subunit *GRIK3*. These can be divided into three main classes: PVALB, SST/GRIK3, and SST/NPY. The PVALB class further splits into a small population that does not express *GRIK3* (the only ones) and a larger PVALB/GRIK3 population, which represents most cells with a high expression of *PVALB*. Intriguingly, the SST/GRIK3 cells are transcriptomically more similar to the *PVALB*+ populations than to the other *SST*-expressing cells, which also express *NPY*. The SST/NPY interneurons, one of the classical groups, can be divided by the presence or absence of *DACH1*, which is highly relevant during human neurodevelopment. In the striatum *DACH1* has been described as co-expressed with *SST* as well as several MSN markers[58].

In the other branch of our classification, we find the two populations representing the most abundant interneuron classes: PTHLH and TAC3. Together with these classes is the well-described cholinergic cells (CHAT), which have already been thoroughly characterized in the literature[3,59–61]. Both PTHLH and TAC3 populations further split into two subclasses of unequal proportions: PTHLH and PTHLH/MOXD1, and TAC3 and TAC3/SEMA3A, respectively.

A *PTHLH*+ population was originally described in the mouse striatum[28], where it was characterized as a group of cells that express *Pvalb* in a gradient manner that correlates with their electrophysiological properties, spatial distribution, morphology, and long-range inputs[31]. In the human striatum, this population is characterized by the expression of *OPN3*, *IL1RAPL2*, and *THSD4*, and is more abundant in the CN than in the Pu. A PTHLH subclass shows a specific expression of *MOXD1*—a monooxygenase predicted to be involved in the dopamine catabolic process—suggesting dopaminergic modulation by these cells. Similarly to the mouse striatum, we find *PVALB* expression in the PTHLH population. This *PTHLH+/PVALB*+ population has also been confirmed by others in the striatum of the human and other species such as the marmoset as well as in the mouse amygdala[37,62]. Interestingly, we have found a specific and less abundant class that expresses *PVALB* (at a significantly higher level than the PTHLH/PVALB cells) but not *PTHLH*. This finding was validated with FISH and differs from the mouse striatum, where all *Pvalb*-expressing cells were also *Pthlh+*[28]. These two classes, PTHLH and PVALB, do not appear close in their molecular identities in the human striatum when applying unbiased hierarchical clustering; however, when we performed a hypothesis-driven analysis of our data, restricted to relevant genes for neuronal functions such as neurotransmitter receptors or ion channels, they showed a very strong correlation. This suggests that even though their overall molecular identities are far apart, these two

classes might share functional roles in the striatal circuit. This observation brings up the recurrent debate on what constitutes a cell class[63–67] and, more importantly, indicates that examining specific aspects of cell identity will deliver different pieces of information, such as what other cell(s) they communicate with (connectivity), what kind of electrical activity they present (electrophysiological properties), the morphological features they exhibit, and/or what is their developmental origin, among others. With that framework in mind, we also analyzed the most relevant genes for FS activity, characteristic of high *Pvalb*-expressing cells in the mouse striatum. Our data showed that in the human striatum, all the FS-relevant genes have high expression levels in both PTHLH and PVALB classes but substantially higher in the latter. Here, we present a comprehensive characterization of the unknown PTHLH class in the human CN and Pu. Importantly, this study confirms their presence in the tissue, even though they had been detected in some of the previous human striatal snRNA-seq datasets[37].

TAC3 was recently described as a primate-specific striatal population[37]. We did define a population of interneurons with high *TAC3* expression as the TAC3 class. However, this class was best defined by the expression of *PTPRK*, *TMEM163*, and *GFRA2*, since *TAC3* is also expressed by the CCK/VIP class, a co-expression that was also observed by Krienen et al.[37]. Through our functional gene analysis, we found that TAC3 interneurons are characterized by synaptic receptors such as glutamate metabotropic (*GRM8*) as well as acetylcholine receptors, both muscarinic (*CHRM2*) and nicotinic (*CHRNA7*, *CHRNB4*). Among the genes with ion channel activity, we found *GRID1* (glutamate ionotropic receptor), *SCN7A* (sodium voltage-gated channel), and *CACNA2D3* (calcium voltage-gated channel). Interestingly, when comparing functional genes in human vs. mouse striatum[28], we found that the mouse interneuron Th cell class had been previously described to express nicotinic receptors, including those responding to a3b4 (a specific subtype)[60,68,69]. This mouse Th population is characterized by the expression of *Tac2* (homologous to the human *TAC3*). Additionally, it expresses *Trh*, one of the best markers for Th cells in mice[28]. Remarkably, both populations share a specific pattern of expression for *PRLR*, *GFRA2* (GDNF receptor), *SYT1*, and *GRIK1*. Notably, neither the mouse Th nor the human TAC3 populations express the last two genes, *SYT1* and *GRIK1*, both of which are involved in synaptic function. Thus, the absence of *SYT1* and *GRIK1* expression is a distinctive feature that distinguishes both human TAC3 interneurons and mouse Th interneurons from all the other interneuron populations in their respective species. They also share the expression of the serotonergic receptor HTR3A/*Htr3a*, which has been used in the cortex as a developmental marker for CGE-derived cells[24]. Interestingly, HTR3A/*Htr3a* is also expressed by the PTLH and CCK/VIP populations in both species. This might indicate a common developmental origin of the PTHLH, TAC3, and CCK/VIP populations.

Although integration of human and mouse datasets was not technically possible, even despite applying recently developed tools such as LIGER[70], the aforementioned genes showed strong parallelism between the mouse Th and the human TAC3 populations. Importantly, in agreement with others[33], we hardly found any *TH*-expressing interneurons, although we did find *TH* expression in MSNs as shown by others[45,71,72] (Supplementary Fig. 9D). This observation points out that *TH* expression in striatum probably cannot be used as a marker for this cell class, at least in humans, and suggests that, from the evolutionary perspective, the absence of *TH* in the TAC3 population might just indicate a refinement in the circuitry or a loss of unnecessary machinery. Noteworthily, since *TH* is the limiting enzyme in the synthesis of dopamine and noradrenaline, we also examined our data for genes related to dopamine metabolism and found none (Supplementary Fig. 11).

Because the discrete partitions of the data might not reveal the entire biologically relevant diversity of the striatal interneuron populations[64], we examined the PTHLH and TAC3 subclasses using factor analysis. With this approach, we did find that there is diversity within the subclasses, as indicated by differences in expression patterns along a continuum rather than by discrete on/off-like changes in the expression of a set of marker genes. The gradients within each subclass were driven by the same set of genes in both the CN and the Pu. We also studied the gradient structure in the human striatum as previously shown in the mouse striatum for both interneuron and MSNs[28–30]. We found an inner gradient structure for the most abundant classes, TAC3 and PTHLH, which is shared in both CN and Pu, indicating similarities in structure, as it was shown for Pthlh interneurons in the mouse striatum[28]. This gradient structure seems to be characteristic of subcortical structures such as the striatum and may reflect the need for a highly specialized organization to receive input from many different and distant brain areas.

The main differences identified between CN and Pu in our study are related to the PTHLH class. Our results indicate that this cell class in the Pu might be involved in long-term potentiation mechanisms, a form of synaptic plasticity that plays a critical role in the proper functionality of the striatum[9,73]. This difference may underscore a potentially different vulnerability of Pu vs. CN to basal ganglia-related diseases, as already suggested by others[74–76], which could be used in the design of cell type-targeted therapeutic approaches.

This study has combined two highly sensitive approaches: snRNAseq and a high-plex in situ platform (Xenium, 10XGenomics) that has demonstrated superior sensitivity compared to other in situ sequencing-based techniques[50,51]

Not only have we established the diversity of interneurons, but we have also demonstrated their presence and prevalence in tissue, which show a high correspondence with the snRNA-seq data, along with their preferred distribution in terms of neighboring cells. Additionally, we performed FISH and presented numerous examples. We further validated our taxonomy by integrating our data with previously published snRNA-seq datasets[37–39]. To accomplish this, we conducted an unbiased clustering of the integrated data, resulting in groups of interneurons that either readily overlapped or were at least related to each of the subclasses we describe here. More relevant, the expression profile of marker genes within each group was consistent across datasets, even in the non-integrated raw data. Our results indicate that our taxonomy is robust, as even the rarest cell subclasses could be observed in tissue and also in other snRNA-seq datasets. Most notably, the classification we introduce was highly compatible with the samples from the nucleus accumbens (ventral striatum) from Tran et al.[38], suggesting that the inhibitory neurons in both ventral and dorsal striatum share a similar diversity spectrum. However, a broader sampling of ventral striatum would be useful to reinforce this observation and determine if this classification can be extended to other regions in the basal ganglia.

In conclusion, we provide a robust, harmonized, transcriptomic-based taxonomy of interneurons in the human striatum with a greater-than-expected diversity of interneuron subclasses and potentially relevant implications for the physiology of the basal ganglia circuit and for the pathophysiology of neurological and psychiatric diseases involving the striatum. Future studies will investigate how these diseases impact the proportions and gene expression profiles of these striatal interneuron's classes and subclasses.

## Methods
### Human tissue
All experiments using post-mortem human tissue were approved by the Swedish Ethical Review Authority (Dnr.: 2020-00341). Pu (*N* = 28) and CN (*N* = 25) fresh frozen tissue samples from 28 donors without neurodegenerative features, aged 25 to over 90 years, were obtained from three sources: the Human Brain and Spinal Fluid Resource Center (Los Angeles, CA, USA), the Parkinson's UK Brain Bank at Imperial (London, UK) and the Massachusetts Alzheimer's Disease Research

Center (MADRC) Neuropathology Core Brain Bank (Charlestown, MA, USA). Donors or their next-of-kin provided written informed consent for brain autopsy, and the study was approved by the review board of each brain bank. Donor information can be found in Supplementary Dataset 1. Given the sex imbalance in our donor sample (9 females and 19 males), our study is underpowered to assess effects or influences due to sex, such as changes in cell class proportions or regulation/ effect on gene expression.

### snRNA-seq data generation

**Tissue dissociation.** The caudate and putamen samples of most donors were obtained from the same flash frozen coronal slab at the level of nucleus accumbens (slab 5 to 7). Brains were cut from the frontal to occipital poles into 5–10 mm thick slabs numbered from 1 to 17–22 depending on the brain size and thickness.

Isolation of nuclei from fresh frozen tissue was performed based on the protocol by the Allen Institute for Brain Science (https://www.protocols.io/view/isolation-of-nuclei-from-adult-human-brain-tissue-eq2lyd1nqlx9/v2) with the following specifications: All steps were performed at 4 °C. 100 – 150 mg of tissue was thawed on ice and homogenized in 2 mL of chilled, nuclease-free homogenization buffer (10 mM Tris pH 8, 250 mM Sucrose, 25 mM KCl, 5 mM $MgCl_2$, 0.1 mM DTT, 1x Protease inhibitor cocktail (50x in 100 % Ethanol, G6521, Promega), 0.2 U/μL Rnasin Plus (N2615, Promega), 0.1 % Triton X-100) using a dounce tissue grinder with loose and tight pestle (20 strokes each, 357538, Wheaton). The nuclei solution was filtered through 70 μm and 30 μm strainers successively, tubes and strainers were washed with an additional homogenization buffer (final volume 6 mL) before centrifugation for 10 min at 900 x $g$. Supernatant was removed leaving 50 μL above the pellet and resuspended in 200 μL homogenization buffer (final volume 250 μL). Then, the suspension was carefully mixed 1:1 with 50% Iodixanol (OptiPrep Density Gradient Medium (D1556, Sigma) in 60 mM Tris (pH 8), 250 mM Sucrose, 150 mM KCl, 30 mM $MgCl_2$) and layered carefully on top of 500 μL 29 % Iodixanol in a 1.5 mL tube. Samples were spun 20 min at 13,500 x $g$ and supernatant was removed as much as possible without disrupting the pellet. Pellet was resuspended in 50 μL chilled, nuclease-free blocking buffer (1x PBS, 1 % BSA, 0.2 U/μL Rnasin Plus), transferred to a fresh tube and filled up to 500 μL. To enable enrichment of neurons during fluorescent-activated nuclei sorting (FANS), 1 μL NeuN antibody (1:500, Millimark mouse anti-NeuN PE conjugated, FCMAB317PE, Merck) was added and samples were incubated for 30 min on ice in the dark. After spinning 5 min at 400 x $g$, the supernatant was removed leaving ~50 μL of buffer above the pellets and 500 μL of blocking buffer was added to resuspend before filtering through a 20 μm filter into FACS tubes and adding 1 μL of DAPI (0.1 mg/mL, D3571, Invitrogen).

**Fluorescent-activated nuclei sorting (FANS).** Nuclei suspension was protected from light and sorted in a flow cytometer (DB FACSAria Fusion or BD FACSAria III) at 4 °C. Gating was performed based on DAPI and phycoerythrin signal into two tubes containing 50 μL blocking buffer (NeuN+ and NeuN- population) until 200,000 nuclei per population were reached. Sorted populations were centrifuged 4 min at 400 x g and supernatant was removed, leaving approximately 30 μL to resuspend the pellet. Samples were kept on ice.

**Library preparation.** Library preparation from sorted nuclei suspension was done using the Chromium Next GEM Single Cell 3' Reagent Kit v3.1 (PN-1000268, 10x Genomics). Each nuclei population was counted manually, and the concentration was adjusted to a range between 200 and 1700 nuclei/μL. Following the manufacturer's protocol (CG000204 Rev D, 10x Genomics), RT mix was added to the nuclei suspension and samples were either loaded for each population on separate lanes (target nucleus recovery 5000) or population were mixed (70% NeuN+ and 30% NeuN, target nucleus recovery 5000 or

7000) before loading on one lane of the Chromium Next GEM Chip G (PN-1000120, 10x Genomics). Downstream cDNA synthesis and library preparation followed the manufacturer's instructions using the Single Index Kit T Set A (PN-1000213, 10 Genomics). Required quality control steps and quantification measurements within this protocol were performed using the Agilent High Sensitivity DNA Kit (5067-4626, Agilent Technologies) and the KAPA Library Quantification Kit (2700098952, Roche).

**Illumina sequencing.** Pools were prepared by combining up to 19 (target nucleus recovery 5000) or 16 (target nucleus recovery 7000) samples and sequencing was performed on a NovaSeq S6000 using a S4-200 (v1.5) flowcell with 8 lanes and a 28-8-0-91 read set up. The sequencing was performed at the National Genomics Infrastructure (Stockholm, Sweden).

### Spatial transcriptomics (Xenium)

We used a high-plex in situ platform (Xenium, 10x Genomics), which reaches subcellular resolution for characterizing RNAs within tissues.

**Gene panel design.** The Xenium technology is based on probes designed to target and detect the expression of a predetermined set of genes, referred to as a panel. The panel used was composed of 266 genes listed in the Xenium Human Brain Gene Expression Panel and 100 additional genes, manually selected based on our snRNA-seq dataset (Xenium Custom Gene Expression Panel 51–100 (Z3DREH), PN-1000561, 10x Genomics).

**Experimental workflow.** Human tissue blocks ($N = 4$, Pu and CN each) were transferred from −80 °C to the cryostat (CryoStar NX70, Thermo Scientific) on dry ice. Samples were mounted on the specimen holder using Tissue Tek O.C.T. Compound (4583, Sakura). Samples and Xenium slides (PN-3000941, 10x Genomics) were acclimated to −20 °C in the cryostat chamber for 5 minutes. A total of 10 μm sections were cut and directly transferred within the imageable area of a chilled Xenium slide (12×24 mm). After positioning the tissue, the slide was briefly warmed from the other side with the thumb to allow the section to adhere and immediately frozen on the cryobar again. Tissue slides were kept in the cryostat during the cutting procedure and stored at −80 °C. Frozen slides were transferred to the in situ Sequencing Infrastructure Unit (Science for Life Laboratory, Stockholm) were probe hybridization, ligation and rolling circle amplification were performed based on the manufacturer's protocol (CG000582 Rev E, 10x Genomics) and background fluorescence was quenched chemically. Using the Xenium Analyser instrument (10x Genomics), the sections were imaged, and the signal was decoded.

### High-sensitivity fluorescent in situ hybridization FISH

**Tissue preparation for histology.** Human tissue blocks ($N = 6$) were stored at −80 °C and transferred to the cryostat (CryoStar NX70, Thermo Scientific) on dry ice. Samples were mounted on the specimen holder using Tissue Tek O.C.T. Compound (4583, Sakura) and acclimated to −20 °C in the cryostat chamber for 5 minutes. 10 μm sections were cut and captured on Super-Frost Plus microscope slides (631-0108, VWR) at room temperature (RT). Slides were air dried at RT for a few minutes and stored for 1 h at −20 °C before transferring them back to −80 °C for long-term storage.

**RNAscope high sensitivity fluorescent in-situ hybridization.** High sensitivity in situ hybridization using the RNAscope Multiplex Fluorescent Reagent Kit v2 (323110, Advanced Cell Diagnostics) was performed on Pu sections of three to six donors to detect single mRNA molecules. Experiments were performed according to the RNAscope Multiplex Fluorescent Reagent Kit v2 protocol (UM 323100, Advanced Cell Diagnostics) for the following genes: *ADARB2* (511651-C3), *CCK*

(539041-C2), *CDH9* (403021-C1), *DACH1* (412041-C1), *DRD1* (524991-C1), *FBXL7* (1200311-C3), *GRIK3* (493981-C1), *NPY* (416671-C2), *OPN3* (1169881-C1), *PTHLH* (452931-C1, 452931-C2), *PVALB* (422181-C2, 422181-C3), *SST* (310591-C3), *TAC3* (507301-C2), *TH* (441651, 441651-C3), *TRH* (409201-C3), *VIP* (452751-C1). In brief, slides were dried at RT for 5–10 min before incubation in 4% PFA for 25 min at 4 °C. Slides were washed twice in 1x PBS and dehydrated in 50%, 70%, and 2×100% ethanol for 5 min each at RT. After drying the slides for 5 min, a hydrophobic barrier was drawn around each section prior to incubation in hydrogen peroxide for 10 min at RT. For antigen accessibility, slides were treated with Protease IV for 20 min at RT after a brief wash in 1x PBS. Slides were washed twice for 3 min again, before probes were incubated. C2 and C3 probes were diluted in C1 probes at a 1:50 ratio and incubated on the slides for 2 h at 40 °C. Slides were then incubated with amplification mix 1–3 followed by a combination of HRP reagent, fluorescent dye and HRP blocker specific for each probe channel in accordance with the manufacturer's recommendations. Probes were detected with Opal fluorophores (Opal 520 (FP1487001, Akoya Biosciences), Opal 570 (FP1488001, Akoya Biosciences), Opal 650 (FP1496001, Akoya Biosciences)) or TSA Vivid dyes (TSA Vivid Fluorophore kit 520 (7523, Tocris), TSA Vivid Fluorophore kit 570 (7526, Tocris), TSA Vivid Fluorophore kit 650 (7527, Tocris). Next, slides were incubated with TrueBlack (23007, Biotium) after a wash in 70% ethanol for 30 sec at RT to quench the autofluorescence due to the accumulation of lipofuscin or other protein aggregates. Prior to mounting with Fluoromount-G (0100-01, SouthernBiotech) the slices, DAPI was added to label the nuclei. A one-day protocol has been used in all experiments to preserve the quality of the slices.

**Image acquisition.** Confocal imaging was performed on a Zeiss LSM800-Airy with Zen software (2.6). 2–3 non-overlapping areas with a size of 8 ×8 tiles were selected per tissue section and images were acquired using a 20x air/dry objective. Final images were stitched using the according feature of the Zen software.

### Quantification and statistical analysis

**Spatial transcriptomics (Xenium) data analysis.** With the aim of identifying the main cell populations present in the datasets generated, nuclear segmentation masks provided in Xenium's output were expanded by 1μm and used to identify the expression of individual cells, generating cell-by-gene expression matrices for all profiled cells across samples. Then, datasets were preprocessed following established best practices[51] (size-based normalization, log-transformation, dimensionality reduction, and graph-based clustering), identifying the main populations present in the datasets. Neuronal clusters were selected based on the expression of *GAD1*, combined with the lack of expression of *OLIG1* and *MOBP*, expressed in oligodendrocytes. Neuronal clusters were further subclustered and clusters corresponding to interneurons were selected based on the expression of either *LHX6* or *SST* and the lack of expression of *MEIS2*.

Interneurons were further annotated into one of the subgroups, previously identified using the snRNA-seq data of this work, based on their positivity on a selection of markers, assigning to one of the subclusters 79% of the annotated interneurons. The remaining interneurons did not exhibit sufficient copies of defined marker genes for classification, reflecting the detection limit constraint of the technique. Cells were then plotted back into the tissue, generating spatial maps. For quantifying the relation between the spatial relation between different cell types, neighborhood enrichment analysis was performed across samples employing Squidpy[77]. Further details can be found in the available code (see the Data and Code availability section below).

**Image analysis for FISH.** Quantitative image analysis was performed using the QuPath software (version 0.3.2[78]) with the following workflow: (1) Definition of region of interest (ROI) on each image across all visible nuclei (using DAPI stain) but excluding artifacts and high fluorescent vessels, based on size and intensity of the signal. (2) Cell and subcellular detection tools of QuPath were adjusted for each staining (Supplementary Dataset 7) and applied within each ROI. (3) For sufficiently good signals a minimum number of subcellular detections were defined as object classifier for a positive cell (*DRD1*). Otherwise, an object classifier was trained for each marker and donor individually by manual labeling of positive cells. Cells were considered positive when either a clear fluorescent signal was visible throughout the approximated cell body (*NPY*, *SST*) or a distinct puncta signal was evident with no overlapping signal in other fluorescent channels (*DACH1*, *TH*, *PTHLH*, *PVALB*). The purpose of using object classifiers instead of counting subcellular spots on tissues with lower signal quality was to improve the distinction between truly positive cells and cells with autofluorescent signal due to lipofuscin. (4) All relevant object classifiers for a specific staining and donor were combined to a composite classifier and applied to all ROI of the respective donor. (5) The resulting list of cells and their assigned markers per ROI were exported and evaluated for each donor within a staining. Among the group of positive cells minimum cut off values concerning the number of subcellular spots were defined for each marker (Supplementary Dataset 7) and applied manually. Additionally, unexpected marker combinations or numbers of spots were checked manually on the image and corrected if necessary.

### snRNA-seq data analysis

**Pre-processing.** The raw snRNA-seq data was processed into count matrices by using CellRanger (v.3.0.0) (10x Genomics) to align the sequencing data to the hg38 genome (GRCh38.p5 (NCBI:GCA_000001405.20), accounting for both intronic and exonic sequences.

**Quality control.** To detect possible doublets, we applied Scrublet[79] to each individual sample 100 times with automated threshold value detection, default parameters, and a random seed. Nuclei labeled as doublets on more than 10% of the Scrublet runs were discarded. Based on the distribution of unique molecular identifiers (UMIs) and unique genes detected per nucleus, cells with less than 500 UMIs or 1200 genes were discarded. Cells with more than 250,000 UMIs, over 15000 genes or more than 10 % mitochondrial content were also excluded. Using the nuclei which passed the initial QC thresholds, we modeled the relationship between number of unique genes and UMIs in the logarithmic scale as a second-degree polynomial function. Nuclei with extreme deviations from the polynomial fit (a difference over 2000 between log(n_genes) and the value predicted by the fit for a given UMI count) were considered outliers and excluded from the rest of the analysis. Cells expressing high levels of marker genes for multiple cell types simultaneously were also discarded. To do so, we computed a cell-type score for each cell type (oligodendrocytes, microglia, oligodendrocyte precursor cells (OPCs), neurons, astrocytes, vascular cells) for each nucleus. This score was the mean expression of the canonical markers for each type. Then we computed the distribution of the scores on the whole dataset, observing bimodal distributions in all cases. We modeled the distributions as mixtures of two Gaussians and set a threshold on the mean of the lowest distribution plus four times its standard deviation. Nuclei with a score above the threshold were considered of a given cell type. Nuclei with scores above the threshold for more than one type were considered doublets and excluded from the rest of the analysis. To remove possible contamination from the claustrum or the amygdala, we removed cells expressing regional markers obtained from the Allen Brain atlas[80]: *NEUROD2*, *TMEM155*, *CARTPT*, *SLC17A7*. The number of nuclei excluded at each step of this process is detailed in Supplementary Fig. 1.

**Interneuron detection**. The count matrices were analyzed using Scanpy[81] to cluster and label them in order to select for interneurons. We performed principal component analysis (PCA) and computed the neighborhood graph on the first 30 principal components (PCs). The data was then clustered using the Louvain algorithm[82] with a resolution of 0.2, and the clusters were labeled as either glia or neurons based on the expression of canonical markers:

Astrocytes – *AQP4, ADGRV1*
Microglia – *CSF1R, FYB1*
Oligodendrocytes – *MBP, MOG, MAG*
Oligodendrocyte precursor cells – *PTPRZ1, PDGFRA, VCAN*
Vascular cells – *EBF1, ABCB1, ABCA9*
Neurons – *MEG3*

Data was visualized in 2-dimensional projections using UMAP, which is a non-linear dimensionality reduction method. In our case, we applied UMAP to the first 30 PCs (30-dimensional) to obtain a 2d representation that could be easily visualized and interpreted. See Ghojogh et al. [83]. for an excellent introduction to the topic. Note that this was employed for visualization only, and all calculations (clustering, correlations) were performed on the high-dimensional data.

The neurons were filtered again based on their distribution of UMIs and genes. Nuclei labeled as neurons with less than 5000 UMIs, less than 3000 genes or more than 12000 genes were discarded, resulting in a total of 181,434 high quality neuronal nuclei. The neurons were re-clustered after removing sex-linked, mitochondrial, and riboprotein genes, projecting them onto their first 30 PCs computed on their 1500 most variable genes. The clusters expressing the inhibitory markers *GAD1* and/or *GAD2* and/or *CHAT* and not expressing MSN (*PPP1R1B, DRD1, DRD2, MEIS2*) or excitatory markers (*RORB*) were labeled as interneurons, resulting in 21,701 nuclei.

*Interneuron classification*. Nuclei labeled as interneurons were projected onto the first 20 PCs calculated on their 1500 most variable genes and re-clustered using the Louvain algorithm. The function rank_genes_groups from Scanpy[81] was used to perform a differential expression analysis between the clusters through a Wilcoxon rank-sum test. We filtered out 2362 nuclei, which formed clusters characterized by low-quality control metrics, excitatory markers (RORB) or MSN markers (*PPP1R1B, DRD1, DRD2, MEIS2*), obtaining a final ensemble of 19,339 high-quality interneuron nuclei. Marker genes were selected manually from the top ranked genes to characterize and name each of the interneuron clusters as a different interneuron subclass. The interneuron subclasses were merged into broader classes based on their correlation. All subclasses with a mean Pearson correlation coefficient higher than 0.49 to each other were joined into a broader class defined by common marker genes. The dendrograms were computed using the average Pearson correlation coefficient between groups across all genes.

*Compositional analysis*. The differences in composition between the CN and the Pu were examined through the centered-log ratio (CLR)[84] values for each interneuron class on each of the regions. This measure is defined as

$$CLR_x = \log\left(\frac{r_x}{g}\right)$$

Where $r_x$ is the fraction of interneurons of a given class and g is the geometric mean of the fractions of each of the classes. The distribution of CLRs for the same interneuron class were compared across regions using a non-parametric Wilcoxon test.

*Differential expression analysis (DEA)*. Regional changes in gene expression were studied using a pseudo-bulk approach in which the nuclei were aggregated by region and sample. The Libra python library[85] was used to perform the data aggregation and the differential expression analysis, which was done using the edgeR-LRT method[86]. In all the other cases, differential gene expression was studied at the cell level using the rank_genes_groups function from the Scanpy library using a Wilcoxon rank-sum test.

*Over-representation analysis (ORA)*. To contextualize the results of the DEA, we analyze the enrichment in specific sets of Gene Ontology terms (known functions, locations and associated processes) on the sets of differentially expressed genes (DEGs). DEGs derived from DEA were used as input into the enrichGO and enrichKEGG functions from the R package clusterProfiler[87]. The p-value is calculated using an hypergeometric test, and p-values are adjusted for multiple comparisons with Benjamini-Hochberg correction. DEGs with a logFC > 0.5 and adjusted p-value < 0.05 were selected. The first function generates functional GO terms[88] related to biological processes, molecular functions, and cellular components. The second function analyzes the enriched terms in our gene list based on the KEGG database[89]. This database is a collection of manually drawn pathway maps representing our knowledge of the molecular interaction, reaction, and relation networks for Metabolism, Genetic Information Processing, Environmental Information Processing, Cellular Processes, Organismal Systems, Human Diseases, and Drug Development. Terms with a p-value < 0.1 were selected as over-represented and plotted using the GOplot package[90].

*Factor analysis*. The heterogeneity within the PTHLH and TAC3 subclasses was studied using a factor analysis. For each interneuron subclass on each of the two striatal regions, we removed the sex-linked, mitochondrial, and riboprotein genes, and then restricted the data to the 1200 most variable genes. We then applied the Factor Analysis function from the scikit-learn Python library[91] with a single latent factor to perform a matrix decomposition.

*Data projection on functional gene subsets*. To restrict the data to neurotransmitter-receptor genes, we selected genes based on their prefixes: *DRD-* (dopamine); *GABR-* (GABA); *CHRN-, CHRM-* (acetylcholine); *GRIA-, GRIN-, GRIK-, GRM-, GRID-, GRIP-* (glutamate). We added three additional glutamate receptors whose naming did not follow the same pattern: *PEPL1, POLR2M*, and *GCOM1*. This selection resulted in 93 genes. To study the genes with ion channel receptors, we restricted our data to the genes listed under the GO-term GO:0005216. This list contained 481 genes names, out of which 431 were found in our data. In both cases, we obtained the UMAP projection from the neighborhood graph computed on the first 30 PCs and then performed a differential expression analysis across interneuron subclasses using a Wilcoxon rank-sum test.

*Public datasets collection and pre-processing*. We collected two single-nuclei RNA-seq datasets of the human striatum from the GEO database[92], with accession numbers GSE151761[37] and GSE152058[39]. A third dataset was obtained from a public repository setup by the authors (https://github.com/LieberInstitute/10xPilot_snRNAseq-human)[38]. On Krienen et al.'s data[37], we analyzed separately the 10x Genomics and Drop-Seq datasets. On Lee et al.'s data[35] we used only the nuclei belonging to control donors (8 samples). We normalized all the datasets using scran normalization[93] and applied individual QC filters to remove bad quality nuclei. We then clustered the data and selected the interneuron populations using the same approach and criteria that we applied to our own data. Notably, on Lee et al.'s dataset our selection included a cluster originally labeled as secretory ependymal cells, which expressed both pan-neuronal and interneuron markers, and we identified as TAC3 interneurons. The total number of cells filtered and selected are detailed in Supplementary Dataset 6.

The mouse data from Muñoz-Manchado et al. [28]. was obtained from the GEO database (accession number GSE97478). The raw counts were normalized using the *normalize_total* function from Scanpy, with a target sum of 10,000 per cell. The original labels were retained, and the data was not transformed further.

*Data integration.* snRNA-seq dataset from multiple sources were integrated using scVI[49]. First, the data was merged and restricted to the 12,986 genes common across datasets. Then the 1200 most variable genes were selected and used to build and train an autoencoder with 1 hidden layer of 128 nodes and a latent space of dimensionality 12 which was trained for 292 epochs. The low-dimensional latent state representation was used to build a neighborhood graph and then cluster the data in the same way as on the PC-projected data from our dataset.

All abbreviations in the text are also indicated in Supplementary Dataset 8.

### Reporting summary
Further information on research design is available in the Nature Portfolio Reporting Summary linked to this article.

## Data availability
Single-cell RNA-seq data have been deposited at Figshare and is available as of the date of publication (https://doi.org/10.6084/m9.figshare.22340140). Xenium in situ data has been deposited at Figshare and Zenodo and is available as of the date of publication (Figshare https://doi.org/10.6084/m9.figshare.25975132; Zenodo https://doi.org/10.5281/zenodo.11609973; https://doi.org/10.5281/zenodo.11534381; https://doi.org/10.5281/zenodo.11612060).

## Code availability
Data analysis code has been deposited at Figshare and is publicly available as of the date of publication. (https://doi.org/10.6084/m9.figshare.22340212). Any additional information required to reanalyze the data reported in this paper is available from the lead contact upon request: Ana B. Muñoz-Manchado (ana.munoz@uca.es).

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

## Acknowledgements

The authors acknowledge the Massachusetts Alzheimer's Disease Research Center (especially Patrick Dooley and Tessa Connors), the Parkinson's UK Brain Bank at Imperial funded by Parkinson's UK (a charity registered in England and Wales, 258197, and in Scotland, SC037554), and the NIH NeuroBioBank (NIH Ref-1851, Requestor: Ernest Arenas), for the supply of tissue samples and associated clinical and neuropathological data. The authors acknowledge support from grants PID2019-109046GA-I00 and PID2022-136526OB-I00 by MCIN/AEI /10.13039/501100011033 and ERDF/EU (A.B.M.-M. and 2019-FPI to J.M.B.-R.), RyC programme RYC-2017-22594 (A.B.M.-M.), the Swedish Foundation for Strategic Research FFL 18-0314 (A.B.M.-M.), the Swedish Research Council 2017-03349 (A.B.M.-M.), the US National Institute on Aging P30AG062421 (B.T.H.) and K08AG064039 (A.S.-P.), the Plan Propio of University of Cádiz (2020-053 / PU / EPIF-FPI-CT to M.D.-S. and Open Access funding to A.B.M.-M). The authors also want to thank the National Genomics Infrastructure in Stockholm funded by Science for Life Laboratory and SNIC/Uppsala Multidisciplinary Center for Advanced Computational Science for assistance with massively parallel sequencing and access to the UPPMAX computational infrastructure. Further thanks go to Katarina Tiklova and Chika Yokota from the In Situ Sequencing Infrastructure Unit at the Science for Life Laboratory, Stockholm. Finally, we are extremely grateful to the brain donors and their families who have made this study possible.

## Author contributions

L.G., L.H. and A.B.M.-M. designed the study; L.H., M.D.-S. and A.B.M.-M. carried out experiments; L.G., J.M.B.-R., L.H., S.M.S. and A.B.M.-M. performed data analysis; M.N., A.S.-P. and B.T.H. provided resources; A.B.M.-M. secured funding; L.G., L.H. and A.B.M.-M. wrote the manuscript with comments and reviewing from all authors.

## Competing interests

S.M.S. is a cofounder of Spatialist AB, a consulting company focused on spatial transcriptomics data analysis. The rest of the authors declare no competing interests.
