## [Peer Review File · Nature Communications]

Interneuron diversity in the human dorsal striatumEditorial Note: This manuscript has been previously reviewed at another journal that is not operating a transparent peer review scheme. This document only contains reviewer comments and rebuttal letters for versions considered at Nature Communications.

REVIEWER COMMENTS

Reviewer #1 (Remarks to the Author):

This is a valuable contribution to increasing understanding of the diversity of interneurons in the human striatum. The topic is important since, as the authors state, striatal interneurons are relevant in several neurological and psychiatric disorders. The research consists in applying RNA sequencing to individual nuclei extracted from postmortem human striatum and then using algorithms to classify them into groups. Synthetically, 14 classes are defined, which can be grouped into 8 main classes. The authors justify these findings with several re-analyses of their data, comparisons with previous data obtained from public repositories, and even from previous animal (rodent) studies. In all, as previously stated, it is a remarkable contribution.

I have some requests and comments to the authors in order to improve the potential application of their results. First of all, it will be useful for all parties (editor and authors) to take into account that my background is in striatal interneurons, although from a very different point of view. As such, my expertise does not allow me to judge the methods used in this research. Given this, however, I think the authors could put more effort into engaging with previous classifications of interneurons coming from neuroanatomy (histochemistry, immunohistochemistry, and so on). For example, the works by Kawaguchi et al ('95 or '98), by Cicchetti, Andre Parent, et al (for example, Brain Research Reviews '00), by Martin Parent (Petryszyn et al, '18), etc, are not cited or discussed throughout the article. This is not a simple demand about 'having to cite this or that', but a request of engaging with previous literature to improve the interpretability of this research. The authors should also consider that, as it is, the manuscript is very hard to understand for someone outside the field of RNA sequencing. Please consider simply the abstract, and keep in mind that the title is as general as "Interneuron diversity in the human dorsal striatum", and therefore will attract a wide diversity of readers. One anatomist approaching it, albeit an expert in striatal interneurons, would be shocked when finding "PTHLH" or "TAC3" as the main groups of interneurons. What does this mean? How does this relate to the beloved cholinergic, calbindin+, parvalbumin+, etc, interneurons? Do these "mouse Th populations" refer to tyrosine-hydroxylase (aka dopaminergic) interneurons of previous works?

This lack of engagement with previous 'anatomical' work (please allow me to polarize between 'anatomical' and 'genetic', for simplicity) is especially evident in the Introduction. For example, the authors state that "the aspiny striatal interneurons have been classically differentiated into two main groups: a small group of cholinergic giant neurons and a diverse population of GABAergic medium-size neurons...". Overall, this is true, but GABAergic interneurons have long been classified according to the calcium-binding protein they express (calbindin, parvalbumin, calretinin). Also, nitroergic interneurons have been classically included (due to the co-expression of neuropeptide Y and somatostatin, they can be termed by those names), as well as the aforementioned tyrosine-hydroxylase interneurons. Also, the authors claim that "striatal interneurons have received little attention compared to the MSNs". Intuitively, this is inaccurate. Trying to justify this intuition, I found in PubMed 302 results searching "striatal interneurons", and 527 hits searching "striatal projection neurons". This does not look like a significant difference. Further, the authors write that previous works have "focused primarily on the cholinergic cells, expressing choline acetyltransferase...", and cite three previous papers. Two of them deal with other interneuronal types apart from cholinergic, and the third article is about cholinergic interneurons indeed, but the same authors have also published on the other interneuronal groups.

Concerning the results, the huge amount of abbreviations (which, if I am not mistaken, are not spelled out in any place of the text) make them difficult to follow. This includes the Figures, which, by the way, are magnificent, but difficult to interpret. For example, not many readers will be familiar with "UMAP projections", which in plain sight seem some sort of anatomical maps of the striatum, but they are not. This may also be the case with "GO-term enrichment analysis". Could these charts be more clearly explained in the legends? Going back to the results, the reader expects some sort of concluding

section (and figure) where the 8 groups and the 14 subgroups of interneurons are clearly summarized. Could it be possible to summarize the meaning (in plain terms) of expressing PTHLH, TAC3, DACH1, etc? Could the authors offer a straightforward visual or narrative characterization of these interneuronal groups? Ideally, following my previous comments, a clear comparison between these "new" groups and the classical "anatomical" groups would be extremely useful. If this is not the case, the reader could end up thinking: ok, this is just a completely different classification of interneurons. So, what should I do now? Do I have to choose between the "anatomical" and the "genetic"? Which one is better? Are they completely unrelated?

With respect to the Discussion, the initial sentence ("The interneuron diversity of the mammalian striatum has received little attention until recently...") seems unjustified considering my comments above, but it is also true that it depends on the point of reference. In the first paragraph, the classical "anatomical" groups are mentioned, although the calbindin+ GABAergic cells are not included. NPY neurons are never referred to as "nitroergic". This is not indispensable, but it could be noted at some point (maybe in this first paragraph, or between lines 400-404). The authors mention some markers for fast-spiking interneurons, but those of tonically-active neurons (TANs, which are considered to be the cholinergic interneurons) are not discussed. Why is this? Are there no genetic markers for these? Maybe these markers would reveal more "cholinergic" interneurons, since their numbers appear to be remarkably lower considering previous reports.

Finally, with respect to the Methods, it would be useful to know which parts of the caudate nucleus and putamen were included. Did the authors resect the whole (at least precommissural and postcommissural regions of the) putamen? What about the caudate? I suppose that the whole nucleus (including the tail) was not resected, but could the authors specify, maybe referring to a human brain atlas, approximately which levels were included? Previous studies have shown that the proportion of interneurons and the presence of the different groups depend on the striatal region, so this could be informative and could explain why the proportion of interneurons (about 11%) is lower than in previous 'anatomical' reports. Also, what about the age of the donor? Could this be an important factor in results? Do the authors keep the data for each brain so they can include age as a covariate, or at least do they have any grounded theory on the (lack of) importance of age?

As a conclusion, I would like to sincerely congratulate the authors on their extremely valuable research. My comments are addressed to improve its understandability and relationship with previous work, so it can reach a wider community of researchers. I hope these comments are useful for such a challenge!

Reviewer #2 (Remarks to the Author):

In this manuscript by Garma and colleagues, the authors investigate the diversity of interneurons in the human dorsal striatum. They use snRNA-seq or postmortem human caudate and putamen and examine the different transcriptional profiles of interneurons.

This is a very important topic, as the role of the striatal interneurons needs much more investigation and, better apprehending the diversity of the different cell types in the striatum, is an essential step. One important strength for this study is the large sampling in comparison to previous studies using tissue from 28 donors which resulted in a bit less than 20,000 nuclei considered as interneurons. After data analysis and clustering, the authors propose 8 main interneurons classes and 14 subclasses based on transcriptomic pattern. While these types of studies are very important for better understanding neuronal diversity in the brain and especially interneurons, I have several major concerns mostly regarding the novelty of the data presented, their consistency with other studies and the conclusions regarding the classifications of interneurons.

Main comments:

- As mentioned above, one great importance of this study relies on the larger sampling in comparison to the other (few) published studies using similar methods. However, the novelty in terms of results and the classification is not evident. Indeed, major clusters (TAC3, SST, ChAT, PTHLH...) were already reported (Krienen et al., 2020, and others) and the differences between the present study and previous ones are not well discussed (see comments below).

- One of the main messages of the manuscript is the potential existence of 14 classes of interneurons selected using snRNA-seq and clustering. However, the justification for some of these 14 subtypes is limited.

o For example, the PTHLH class was here subdivided based on the expression of MOXD1. What is the rationale for selecting these 2 subclasses? It seems that some PTHLH cells would also express PV. Why suggesting a PTHLH/MOXD1 population but not a PTHLH/PV?

o The authors suggest the existence of 4 classes of CCK interneurons based on the expression of VIP (another "classic" marker used for interneurons classification) but also other markers that are not defined such as CHST9 or CXCL14. Again, what is the main rationale for subdividing these CCK cells into 4 populations? Why not one population of ADARB2+ or two populations of CCK and CCK/VIP? Especially as in the discussion (line 394) CCK-expressing cells are then described altogether.

o The TAC3 population is subdivided in TAC3 and TAC3/Sema3A. But in the discussion, it is mentioned that TAC3 is also expressed by the CCK/VIP class. Why was it not considered as a separate subclass? Or why not considering grouping all TAC3 cells together?

- Some results presented here seem to contradict some pre-existing studies performed in primates. Here are a few examples:

o Previous studies report a much larger population of PV+ interneurons in the striatum and strong overlap with PTHLH.

o Several studies including ones performed in primates, describe a population of THINs. The discrepancies (as well as others) between these studies and previous ones should be better discussed.

- On the same lines, the author should acknowledge much more other studies that have investigated different cell types in the striatum both in rodents as well as in primates. While some of these were obtained using different techniques, they provided very important information regarding the classification of GABAergic interneurons, their role in striatal circuits and in behavior. These past studies are cited (for a minority of them) but not used in the context of the classification of different populations of interneurons, the differences observed between studies, the potential implications...

- The validation of the proposed interneuron classification using FISH is too superficial and should be extended including more markers. Further, it should be discussed whether the quantifications obtained with both techniques reach similar results. Finally, this has been done mostly for established markers such as SST, NPY, PV. Importantly, FISH of less-established markers such as DACH1 seems to give contradictory results with the snRNA-seq, which brings several questions regarding the validity of the classification.

- The number of groups defined in different parts of the study is a bit confusing. The main claim is that there could be 14 subtypes, but when investigating the regional differences this was done only in 8 populations. Then, when looking at functionally relevant genes this was observed for 7 populations (CCK, CCK/VIP, PV, SST/GRIK3, PTHLH, ChAT, TAC3). Finally, when including data from other studies, they found 16 groups. This lack of consistency makes the proposed subdivision more confusing.

- The process to select interneurons should be better described.

o It says that MSNs were discarded based on D1 or D2 R expression. Are the authors certain that this will not exclude some interneuron populations?

o Excitatory neurons were excluded. If samples were taken from the striatum, what are the sources of excitatory neurons?

o These are selected based on the expression of GAD1 and/or GAD2. Wouldn't this exclude the majority of ChAT+ interneurons?

- One more general comment is the definition of a cell type. As mentioned here in lines 423-427:

"These two classes, PTHLH and PVALB, do not appear close in their molecular identities in the human striatum when applying unbiased hierarchical clustering; however, when we performed a hypothesis-driven analysis of our data, restricted to relevant genes for neuronal functions such as neurotransmitter receptors or ion channels, these two classes showed a very strong correlation." Depending on the analysis methods (or different studies using similar approaches) "cell types" can be considered as unique or different especially considering some gradient expression of some genes. Further, as mentioned above this study doesn't acknowledge enough other criteria (at least as important) used to define a cell type (anatomy, connectivity, electrophysiological properties...) and the research that has been done (in rodents, monkeys, humans...) to classify these cell types, the differences with the current study and potential hypotheses.

This last fact is also reflected in some comments in the introduction:

- Line 56: "The aspiny striatal interneurons". Several studies have shown that several populations of interneurons are sparsely spiny.

- Line 57: "small group of cholinergic giant neurons". In the majority of studies, cholinergic interneurons are not described as a minor group representing 1-2% of striatal neurons which is more important than several populations of GABAergic interneurons.

- Lines 59-67: "Since the striatal interneurons have received little attention compared to the MSNs, consensus regarding the populations comprising these neuronal groups and how to identify them is lacking. However, recent advances such as new transgenic reporter mice that target the complete striatal and cortical interneuron repertoire^{15,16}, and single cell/nucleus RNA-sequencing (sc/nRNA-seq) have enabled large-scale approaches to investigate cell diversity based on the individual cell transcriptome^{17,18,19} in different mouse brain areas including the striatum ^{20,21,22}. Using these methods, a recent study identified seven interneuron populations in the mouse striatum based on their molecular and electrophysiological profile: Npy/Sst, Npy/Mia, Cck/Vip, Cck, Chat, Th, and Pthlh ²⁰" What about all the other evidence obtained before and after the introduction of transgenic mice using methods different than sc/snRNAseq? All the original studies describing the existence of SST, NPY, NGFs, THINs, PV+-FSIs, 5HT3a... are not cited and not accounted for.

- Lines 74-79: Most of the studies are limited by the technical approach because they have relied on the classical markers to identify interneuron populations and focused primarily on the cholinergic cells, expressing choline acetyltransferase (ChAT)^{24,25,26}. Prior snRNA-seq studies on the human and non-human primate striatum have highlighted different aspects, such as broad differences across species and brain areas^{27,28} or in health vs. disease²⁹, but lack sufficient interneuron sampling to characterize striatal interneuron diversity."

This is not true on several levels. 1) Classification of interneurons did not start with snRNAseq and 2) multiple studies have investigated interneuron diversity without only focusing on cholinergic interneurons. These are just not cited here.

Manuscript NCOMMS-23-21576-T, Garma *et al.*

RESPONSE TO REVIEWERS POINT BY POINT

We are grateful to the reviewers for their thoughtful feedback and constructive suggestions.

Reviewer 1 (Remarks to the Author): *This is a valuable contribution to increasing understanding of the diversity of interneurons in the human striatum. The topic is important since, as the authors state, striatal interneurons are relevant in several neurological and psychiatric disorders. The research consists in applying RNA sequencing to individual nuclei extracted from postmortem human striatum and then using algorithms to classify them into groups. Synthetically, 14 classes are defined, which can be grouped into 8 main classes. The authors justify these findings with several re-analyses of their data, comparisons with previous data obtained from public repositories, and even from previous animal (rodent) studies. In all, as previously stated, it is a remarkable contribution.*

Answer #1:

We thank the reviewer for his/her comments. We find that he/she provided useful constructive criticism and we have followed it to improve the quality of our manuscript.

Reviewer 1: *I have some requests and comments to the authors in order to improve the potential application of their results. First of all, it will be useful for all parties (editor and authors) to take into account that my background is in striatal interneurons, although from a very different point of view. As such, my expertise does not allow me to judge the methods used in this research. Given this, however, I think the authors could put more effort into engaging with previous classifications of interneurons coming from neuroanatomy (histochemistry, immunohistochemistry, and so on). For example, the works by Kawaguchi *et al* ('95 or '98), by Cicchetti, Andre Parent, *et al* (for example, *Brain Research Reviews* '00), by Martin Parent (Petryszyn *et al*, '18), etc, are not cited or discussed throughout the article. This is not a simple demand about 'having to cite this or that', but a request of engaging with previous literature to improve the interpretability of this research.*

Answer #2:

We thank the reviewer for this comment. We have thoroughly reviewed the introduction section to expand on the existing classification of interneurons and also added the following citations describing the classical taxonomy: Kawaguchi Y *et al.* 1995 (ref. 16), Cicchetti F *et al.* 2000 (ref. 17), Petryszyn *et al.* 2018 (ref. 19), Cossette *et al.* 2003, Lallani S.B. *et al.* 2019 (red. 33), Araújo de Góis *et al.* 2023 (ref. 34). We had already cited as a reference a recent review from Tepper *et al.* 2018 (ref 15), which includes works from (Kawaguchi *et al* 1995 and 1998) and Martin Parent (Petryszyn S., *et al.* 2016) but have now included those seminal articles as well.

Reviewer 1: *The authors should also consider that, as it is, the manuscript is very hard to understand for someone outside the field of RNA sequencing. Please consider simply the abstract, and keep in mind that the title is as general as "Interneuron diversity in the human dorsal striatum", and therefore will attract a wide diversity of readers. One anatomist approaching it, albeit an expert in striatal interneurons, would be shocked when finding "PTHLH" or "TAC3" as the main groups of interneurons. What does this mean? How does*

this relate to the beloved cholinergic, calbindin+, parvalbumin+, etc, interneurons? Do these “mouse Th populations” refer to tyrosine-hydroxylase (aka dopaminergic) interneurons of previous works?

Answer #3:

We agree that we have indeed aimed our article at a general audience and we could make it significantly more accessible to readers outside the single-cell transcriptomic field. We have now simplified the abstract and elaborated on how the “classical” markers PVALB, CALB1 (Calbindin), CALB2 (Calretinin), TH, NOS1, SST, NPY and CHAT map onto our new taxonomy (see Figure 1E and Supplementary Figure 3, and Results Section). In summary, we investigated classical markers (PVALB, CALB1, CALB2, NOS1, SST, NPY, TH, and CHAT) previously utilized for defining striatal interneuron diversity. Several markers were expressed across various classes in our taxonomy. For example, CALB2 (Calretinin) was highly expressed in the CCK/VIP class and lower in the TAC3 class. CALB1 was present in the CCK/CHST9 and SST/NPY/DACH1 subclasses. Classical nitrenergic cells expressing NOS1 were found in the SST/NPY and SST/NPY/DACH1 subclasses. PVALB and PVALB/GRIK3 subclasses represented parvalbumin-expressing cells, and the CHAT class denoted cholinergic cells. Overall, the snRNA-seq data expanded the classical division, enabling further subclassification based on complete transcriptomic profiles. Notably, the PTHLH population lacked characterization by classical markers but exhibited low *PVALB* and sparse *TH* expression as recently shown in the mouse striatum. Classical markers unique to TAC3 cells were not detected, and TH expression in interneurons remained negligible, consistent with previous snRNA-seq studies. SST/GRIK3 interneurons expressed SST but did not co-express NOS1 and NPY, typical of SST+ interneurons. Instead, they expressed GRIK3, TAC1, and TAC3. Thus, the PTHLH and SST/GRIK3 classes did not align with any classical markers, and TAC3 expressed CALB2 at low level. Concerning TH-expressing cells in the human striatum and the TAC3 class's identity, extensive discussion was provided in Figure 5, and now we have further extended our effort including Supplementary Figure 9. In summary, our findings suggest that TH is not a significant interneuron marker in the human striatum but is predominantly expressed by MSNs, which is also in accordance with other studies. Interestingly, there are relevant genetic similarities between the human TAC3 and the mouse Th population, indicating potential equivalency, and suggesting a TH expression lost probably due to evolutionary reasons not explored in this study.

Reviewer 1: This lack of engagement with previous ‘anatomical’ work (please allow me to polarize between ‘anatomical’ and ‘genetic’, for simplicity) is especially evident in the Introduction. For example, the authors state that “the aspiny striatal interneurons have been classically differentiated into two main groups: a small group of cholinergic giant neurons and a diverse population of GABAergic medium-size neurons...”. Overall, this is true, but GABAergic interneurons have long been classified according to the calcium-binding protein they express (calbindin, parvalbumin, calretinin).

Answer #4:

We thank the reviewer for this comment. As mentioned above, we have now revised the Introduction section to properly cite the classification of GABAergic interneurons according to the expression of Calbindin, Parvalbumin, Calretinin among other classical markers (page 2, text in blue). We have also added the expression of these calcium binding protein genes used as classical markers onto our transcriptomic-based classification (Figure 1E), and extensively analyzed them in the Results and Discussion sections.

Reviewer 1: Also, nitrergic interneurons have been classically included (due to the co-expression of neuropeptide Y and somatostatin, they can be termed by those names), as well as the aforementioned tyrosine-hydroxylase interneurons.

Answer #5:

We appreciate this reviewer's comment. As indicated in answer #3, we have now included the correspondence of the nitrergic neurons to the taxonomy we present here (Figure 1E, Supplementary Figure 3) and we have also extensively discussed the TH interneurons (Figure 5 and Supplementary Figure 9).

Reviewer 1: Also, the authors claim that “striatal interneurons have received little attention compared to the MSNs”. Intuitively, this is inaccurate. Trying to justify this intuition, I found in PubMed 302 results searching “striatal interneurons”, and 527 hits searching “striatal projection neurons”. This does not look like a significant difference.

Answer #6:

We agree with this reviewer that there is not enough evidence to substantiate this claim and have now removed this sentence from the Introduction.

Reviewer 1: Further, the authors write that previous works have “focused primarily on the cholinergic cells, expressing choline acetyltransferase...”, and cite three previous papers. Two of them deal with other interneuronal types apart from cholinergic, and the third article is about cholinergic interneurons indeed, but the same authors have also published on the other interneuronal groups.

Answer #7:

We thank the reviewer for noting these wrong citations. We have now removed that phrase.

Reviewer 1: Concerning the results, the huge amount of abbreviations (which, if I am not mistaken, are not spelled out in any place of the text) make them difficult to follow. This includes the Figures, which, by the way, are magnificent, but difficult to interpret. For example, not many readers will be familiar with “UMAP projections”, which in plain sight seem some sort of anatomical maps of the striatum, but they are not. This may also be the case with “GO-term enrichment analysis”. Could these charts be more clearly explained in the legends?

Answer #8:

We have added Supplementary Table 8, which contains a comprehensive list of abbreviations used in the text. Additionally, we have provided the full spellings for each abbreviation within

the manuscript. In the methods section, we have included a brief description of both UMAP and GO-term enrichment:

UMAP: “Data was visualized in 2-dimensional projections using UMAP, which is a non-linear dimensionality reduction method. In our case, we applied UMAP to the first 30 PCs (30-dimensional) to obtain a 2d representation that could be easily visualized and interpreted. See Ghojogh et al. (Ghojogh, Benyamin, et al. "Uniform Manifold approximation and projection (UMAP) and its variants: tutorial and survey." *arXiv preprint arXiv:2109.02508* (2021).) for an excellent introduction to the topic. Note that this was employed for visualization only, and all calculations (clustering, correlations) were performed on the high-dimensional data.”

GO-TERM: “To contextualize the results of the DEA, we analyze the enrichment in specific sets of Gene Ontology terms (known functions, locations and associated processes) on the sets of differentially expressed genes. Differentially Expressed Genes (DEGs) derived from DEA were used as input into the *enrichGO* and *enrichKEGG* functions from the R package *clusterProfiler*⁶⁸. DEGs with a $\log_{2}FC > 0.5$ and p -adjusted value < 0.05 were selected. The first function generates functional GO terms⁶⁹ related to biological processes, molecular function, and cellular components. The second function analyzes the enriched terms in our gene list based on the KEGG database⁷⁰. This database is a collection of manually drawn pathway maps representing our knowledge of the molecular interaction, reaction, and relation networks for Metabolism, Genetic Information Processing, Environmental Information Processing, Cellular Processes, Organismal Systems, Human Diseases, and Drug Development. Terms with a p -value < 0.1 were selected as over-represented and plotted using the *GOplot* package⁷¹.”

Reviewer 1: Going back to the results, the reader expects some sort of concluding section (and figure) where the 8 groups and the 14 subgroups of interneurons are clearly summarized. Could it be possible to summarize the meaning (in plain terms) of expressing PTHLH, TAC3, DACH1, etc? Could the authors offer a straightforward visual or narrative characterization of these interneuronal groups? Ideally, following my previous comments, a clear comparison between these “new” groups and the classical “anatomical” groups would be extremely useful. If this is not the case, the reader could end up thinking: ok, this is just a completely different classification of interneurons. So, what should I do now? Do I have to choose between the “anatomical” and the “genetic”? Which one is better? Are they completely unrelated?

Answer #9:

We thank the reviewer for this comment. The names of the IN classes and subclasses are chosen after main marker genes as it is usually done. In some cases (e.g. TAC3), we did not pick the best marker gene in order to maintain consistency with previous classifications based on transcriptomics and explained that in the text. As per your thoughtful suggestion, we have added classical markers expression in Figure 1E and Supplementary Figure 3 and commented on it in the results section to relate our taxonomy to the neuroanatomical classical division. As we show now, we believe these two classifications are complementary based on our mapping of our interneuron subclasses and the classical anatomical types (Calretinin – CCK/VIP, CCK/VIP/CXCL14, TAC3, and TAC3/SEMA3A; Calbindin – CCK/CHST9 and SST/NPY/DACH1; Nitroergic – SST/NPY and SST/NPY/DACH1; PV+ - PVALB and PVALB/GRIK3). However the snRNA-seq approach provides more granularity within each main class by revealing transcriptomic-based subclasses, which may have relevant functional correlates as our analyses suggest. Besides we reveal cell classes not detected with classical markers as PTHLH and SST/GRIK3. And discuss the identity of TAC3 class (see answer #3).

Reviewer 1: With respect to the Discussion, the initial sentence (“The interneuron diversity of the mammalian striatum has received little attention until recently...”) seems unjustified considering my comments above, but it is also true that it depends on the point of reference. In the first paragraph, the classical “anatomical” groups are mentioned, although the calbindin+ GABAergic cells are not included. NPY neurons are never referred to as “nitrgergic”. This is not indispensable, but it could be noted at some point (maybe in this first paragraph, or between lines 400-404). The authors mention some markers for fast-spiking interneurons, but those of tonically-active neurons (TANs, which are considered to be the cholinergic interneurons) are not discussed. Why is this? Are there no genetic markers for these? Maybe these markers would reveal more “cholinergic” interneurons, since their numbers appear to be remarkably lower considering previous reports.

Answer #10:

We acknowledge this reviewer's concern regarding the first sentence of the Discussion section. The sentence was not intended to devalue previous works but rather to emphasize the significance of the present study. We concur with this reviewer that interpretation can be influenced by the chosen reference point, and, after careful consideration, we have opted to retain the sentence. Regarding the classical classification issues, as explained in previous answers we have now analyzed the correspondence of our transcriptomic-based taxonomy with the classical anatomical classification and included the term nitrgergic in the text. Regarding the cholinergic interneurons, usually referred to also as “Chat cells,” we clearly identified them and kept the same nomenclature, as *CHAT* is one of the main marker genes, together with *SLC5A7*. We did not discuss them further as there was a clear agreement with previous classifications. The low number of cholinergic cells in the snRNA-seq data might be justified for several reasons: (1) we are covering a specific level of putamen and caudate where they are less abundant; (2) our detection reveals abundant interneuron groups (PTHLH and TAC3) that were not fully considered in previous anatomical studies, thus, when factoring in these additional interneuron groups, the proportion of cholinergic cells appears considerably lower and, perhaps the most likely explanation, (3) there is a methodological bias inherent to the snRNA-seq technique since the spatial transcriptomic approach did reveal a 13% of CHAT interneurons, which matches perfectly with other previous anatomical studies (e.g., 13,7% in Lecumberri *et al.* “Neuronal density and proportion of interneurons in the associative, sensorimotor and limbic human striatum”. *Brain Structure and Function* 223 (2018): 1615-1625).

Reviewer 1: Finally, with respect to the Methods, it would be useful to know which parts of the caudate nucleus and putamen were included. Did the authors resect the whole (at least precommissural and postcommissural regions of the) putamen? What about the caudate? I suppose that the whole nucleus (including the tail) was not resected, but could the authors specify, maybe referring to a human brain atlas, approximately which levels were included? Previous studies have shown that the proportion of interneurons and the presence of the different groups depend on the striatal region, so this could be informative and could explain why the proportion of interneurons (about 11%) is lower than in previous ‘anatomical’ reports.

Answer #11: We appreciate this reviewer’s question. Indeed, we would expect some compositional variations along the antero-posterior axis of both caudate and putamen as the anatomical studies highlight. The caudate and putamen samples of most donors were obtained from the same flash-frozen coronal slab at the level of the nucleus accumbens (slab 5 to 7).

Brains were cut from the frontal to occipital poles into 5-10 mm thick slabs numbered from 1 up to 17-22 depending on the brain size and slab thickness. We have now added this information to the Methods Section. Our underestimation of *CHAT*+ cells is the most likely explanation for the lower interneuron detection.

Reviewer 1: Also, what about the age of the donor? Could this be an important factor in results? Do the authors keep the data for each brain so they can include age as a covariate, or at least do they have any grounded theory on the (lack of) importance of age?

Answer #12:

Unfortunately, the sampling in this study does not allow us to perform a reliable correlation between the abundance of each interneuron subtype and the age of the donors treated as a continuous variable, as our dataset is inevitably enriched in older age donors because of natural reasons concerning availability of postmortem tissue. However, we have included a supplementary figure (Supplementary figure 4) to show that all subtypes are present in all age groups with age binned in <50, 50-70, 70-90, and >90 intervals. As we show, all cell classes are represented in all age groups; we have commented on this in the Results section (last paragraph):

”Regarding the consistency of the classes along age we observed that all interneuron subclasses were present in samples from all the age groups in our dataset (Supplementary Figure 4).”

Reviewer 1: As a conclusion, I would like to sincerely congratulate the authors on their extremely valuable research. My comments are addressed to improve its understandability and relationship with previous work, so it can reach a wider community of researchers. I hope these comments are useful for such a challenge!

Answer #13:

We would like to thank the reviewer once again for the encouraging remarks and for very insightful comments. We believe that they have been very useful to improving our manuscript and making it accessible to a wider audience and connecting our study with previous works.

Reviewer #2: *(Remarks to the Author): In this manuscript by Garma and colleagues, the authors investigate the diversity of interneurons in the human dorsal striatum. They use snRNA-seq or postmortem human caudate and putamen and examine the different transcriptional profiles of interneurons.*

This is a very important topic, as the role of the striatal interneurons needs much more investigation and, better apprehending the diversity of the different cell types in the striatum, is an essential step.

One important strength for this study is the large sampling in comparison to previous studies using tissue from 28 donors which resulted in a bit less than 20,000 nuclei considered as interneurons.

After data analysis and clustering, the authors propose 8 main interneurons classes and 14 subclasses based on transcriptomic pattern. While these types of studies are very important for better understanding neuronal diversity in the brain and especially interneurons, I have several major concerns mostly regarding the novelty of the data presented, their consistency with other studies and the conclusions regarding the classifications of interneurons.

Answer #14:

We thank the reviewer for their valuable feedback and insightful comments. We have revised our manuscript following their specific suggestions to clarify the novelty of our work and its relation with previous studies as well as our methods.

Reviewer 2: As mentioned above, one great importance of this study relies on the larger sampling in comparison to the other (few) published studies using similar methods. However, the novelty in terms of results and the classification is not evident. Indeed, major clusters (TAC3, SST, ChAT, PTHLH...) were already reported (Krienen et al., 2020, and others) and the differences between the present study and previous ones are not well discussed (see comments below).

Answer #15:

We value the reviewer's concerns. Currently, our work stands as the sole dedicated snRNA-seq study aimed at unraveling the diversity of interneurons in the human caudate nucleus and putamen, making it a novel contribution. Regarding the other three snRNA-seq studies that contain any striatal interneurons, as highlighted by the reviewer, they provide limited information considering the low number of nuclei sequenced and donors included, and that none encompassed both the caudate nucleus and putamen). Moreover, they yielded lower quality data due to a shallower sequencing depth. Additionally, the absence of consistency in their classifications results in highly divergent taxonomies, leaving the issue of interneuron diversity in the human striatum unsolved. More details about each specific study follows below:

- 1) The snRNA-seq study from **Lee et al.** does not provide any comment on the human striatal interneurons, but their figures show 3 clusters labeled as interneurons: PVALB/TH, SST/NPY, and CHAT, with no mention of other types (PTHLH, TAC3...). Remarkably, their data revealed that the nuclei labeled as “Secretory

ependymal” and as “FOXP2 Neurons” by the authors contain an abundance of nuclei expressing GABAergic markers GAD1 and GAD2:

- 2) The classification from **Tran *et al.*** is from nucleus accumbens (i.e., neither caudate nor putamen), and describes five types of interneurons, named A to E by the authors. They do not present a clear relation of these types with other classifications and only mention the prevalence of several known marker genes: *SST*, *NPY* - E; *VIP* - B; *TAC3* - A. Classes C and D are claimed to “likely represent unique PV-expressing interneurons” although the authors state that “we did not observe robust expression of parvalbumin (*PVALB*) in any cluster”. Notably, this classification does not include cholinergic interneurons, as the authors were unable to identify them. Similarly, no group of neurons was identified as *TH*⁺.
- 3) **Krienen *et al.*** present a broad study focusing on interneuron variations across five different species in cortical and subcortical areas. However, their examination of human striatal results is limited to only five donors and the methods section lacks specificity regarding the inclusion of specific parts of the human striatum, referencing only the caudate nucleus in relation to interneuron diversity from humans (see extended figure 10). Notably, their interneuron classification in the human striatum reveals seven distinct classes (*SST*⁺, *CHAT*⁺, *ADARB2*⁺, *MEIS2*⁺, *TH*⁺, *PTHLH*⁺, and *TAC3*⁺),

none of which aligns with our classification, except for cholinergic interneurons (see Supplementary figure 10). It is crucial to highlight that *MEIS2*⁺ is entirely absent in the figures depicting human donors. The discrepancy further extends to their online tool (<http://interneuron.mccarrolllab.org>), which lacks a PTHLH population, introduces *PVALB*⁺, and substitutes *ADARB2*⁺ with *CCK*⁺. Consequently, their classification lacks clarity, as reflected in the tool's displayed classes: *CCK*⁺, *CHAT*⁺, *SST*⁺, *TH*⁺, *PVALB*⁺, and *TAC3*⁺. Regarding the TH population, while they designate a group as *TH*⁺, they fail to detect *TH* expression in human interneurons. Consequently, the gene expression pattern defining the *TH*⁺ group in humans, along with the novel PTHLH and TAC3 populations, remains undefined in their study. In contrast, we identify *TAC3*⁺ expressing cells and argue that *PTPRK* serves as the most accurate marker for the TAC3 population, as TAC3 is also expressed by *CCK*-expressing cells. To maintain clarity and simplicity in the field, we deliberately chose not to introduce additional "cluster names." Instead, we opted for a comprehensive approach, conducting an exhaustive analysis. Our findings, illustrated in Figure 5, reveal that the TAC3 population we define is analogous to the Th population in mice. This significant correlation is thoroughly explored and discussed in our manuscript. This significant finding differs markedly from Krienen et al., who classify *TAC3*⁺ expressing cells as an entirely unrevealing group of interneurons. Given the extensive discussion surrounding TH interneurons in the field over the years, our statement contributes substantially to the current discourse. Regarding PTHLH, while Krienen et al. merely establish the presence of *PTHLH* expression in their study, we conduct a comprehensive analysis and draw comparisons with the mouse striatum, where we initially defined this population.

In summary, despite the existence of three prior snRNA-seq studies on human striatal interneurons, as referenced in our manuscript, their divergent focus, limited sampling, variation in dorsal striatal areas covered if any, and shallower sequencing have resulted in a taxonomy that is neither clear nor consistent. In contrast, our study undertakes a profound analysis with a substantial investment in sampling, coverage, sequencing methodology, and computational analysis with a particular emphasis on exploring novel populations. Following the reviewer's comments, we have now extended our effort to establish a meaningful correspondence with classical taxonomies from previous studies including classical markers expression in Figure 1D-E. The overarching goal of our research is to comprehensively define human interneuron diversity, marking a significant stride in this field. In order to highlight the significance of our work, we have further elaborated on our claim regarding the novelty of the present study:

“Our sampling comprises nearly half a million nuclei overall, which constitutes by far the largest study of this kind to date and the first one to robustly identify distinct groups using two highly sensitive techniques, such as snRNAseq and spatial transcriptomics.”

Reviewer 2: One of the main messages of the manuscript is the potential existence of 14 classes of interneurons selected using snRNA-seq and clustering. However, the justification for some of these 14 subtypes is limited.

Answer #16:

We thank the reviewer for their comment. We did not choose the number of subclasses, but rather we followed a data-driven approach: the methods section describes that the cells were clustered, we performed differential expression among clusters and then “Marker genes were selected manually from the top ranked genes to characterize and name each of the interneuron clusters as a different interneuron subclass.”. This is the standard approach used to analyze single-cell/single-nuclei data (see guidelines for bioinformatics of single-cell sequencing data PMID: 35236372, and study examples Yang, Chao, et al. "Heterogeneity of human bone marrow and blood natural killer cells defined by single-cell transcriptome." *Nature communications* 10.1 (2019): 3931., Grubman, Alexandra, et al. "A single-cell atlas of the entorhinal cortex from individuals with Alzheimer’s disease reveals cell-type-specific gene expression regulation." *Nature neuroscience* 22.12 (2019): 2087-2097., Park, Jihwan, et al. "Single-cell transcriptomics of the mouse kidney reveals potential cellular targets of kidney disease." *Science* 360.6390 (2018): 758-763., Velmishev, Dmitry, et al. "Single-cell genomics identifies cell type-specific molecular changes in autism." *Science* 364.6441 (2019): 685-689.). When picking marker genes, we aimed at selecting genes that were as exclusive as possible, and we intended to use a nomenclature that related to previous works (i.e. we used the names TAC3 and PTHLH although these genes are not the most distinctive in the respective classes to maintain consistency with previous studies).

The full list of differentially expressed genes was provided in the supplementary materials (Supplementary table 2) to enable readers to examine all genes on each class and subclass.

We have rephrased the text in the Results section to include a brief description of the method that we employed:

“We clustered all the interneurons, resulting in 14 clusters which we identified as 14 different interneuron subclasses based on the expression of unique transcriptomic patterns. Merging highly correlated classes (see Methods), we produced a broader classification with eight main classes, which we named after selected marker genes: CCK/VIP (*ADARB2+*, *CCK+*, and *VIP+*), CCK (*ADARB2+* and *CCK+*), PVALB (*PVALB+*), SST/GRIK3 (*SST+* and *GRIK3+*), SST/NPY (*SST+* and *NPY+*), PTHLH (*PTHLH+* and *OPN3+*), CHAT (*CHAT+* and *SLC5A7+*) and TAC3 (*TAC3+* and *PTPRK+*). The nuclei assigned to different classes and subclasses can be seen separated from each other when projected on the 2-dimensional uniform manifold approximation (UMAP) (Figure 1D, E). The main transcriptomic patterns that distinguish each subclass are shown in Figure 1F, whereas the complete results of a differential expression analysis at class and subclass levels are provided in Supplementary table 2.”

We address the specific reviewer’s suggestions regarding cell types below in the next answers (#17 to #20).

Reviewer 2: For example, the PTHLH class was subdivided here based on the expression of MOXD1. What is the rationale for selecting these 2 subclasses? It seems that some PTHLH cells would also express PV. Why suggesting a PTHLH/MOXD1 population but not a PTHLH/PV?

Answer #17:

As elaborated in the previous response, it is important to note that the clustering process is not arbitrary; rather, it follows a well-defined pipeline approach commonly employed in the scRNA-seq field. Two distinct clusters exhibited high expression of the PTHLH markers: *OPN3*, *THSD4*, and *PTHLH* itself. The differentiation between these two clusters was

discerned based on the expression of *MOXD1*. Conversely, the presence of *PVALB* in the PTHLH population was characterized by both low and sparse expression and did not serve as a distinguishing factor between the two clusters exhibiting high levels of PTHLH markers. To visually elucidate this point, we have included here a figure displaying the expression patterns of *PTHLH*, *MOXD1*, and *PVALB*.

Reviewer 2: *The authors suggest the existence of 4 classes of CCK interneurons based on the expression of VIP (another “classic” marker used for interneurons classification) but also other markers that are not defined such as CHST9 or CXCL14. Again, what is the main rationale for subdividing these CCK cells into 4 populations? Why not one population of ADARB2+ or two populations of CCK and CCK/VIP? Especially as in the discussion (line 394) CCK-expressing cells are then described altogether.*

Answer #18:

Having previously explained the rationale behind our clustering approach in earlier responses, let us focus on the specific context of the four classes of CCK interneurons. As detailed in the Results section and illustrated in Figure 1F, we find four *ADARB2*⁺ clusters that segregate according to the expression of markers *CCK* and *VIP*. We found that the expression of *CHST9* and *CXCL14* could distinguish the two *ADARB2*⁺/*CCK*⁺ clusters (Figure 1F). We include here a plot showing the expression of *ADARB2*, *CCK*, *VIP*, *CHST9*, and *CXCL14* to further clarify this point:

Because we did find four *ADARB2*⁺ clusters and because we could characterize them by their expression pattern, we decided to maintain our classification as it is, even though it uses novel marker genes such as *CHST9* and *CXCL14*. We did merge these four subclasses into two main ones, CCK/VIP and CCK, because they were highly correlated (as detailed in the Methods section: “The interneuron subclasses were merged into broader classes based on their correlation. All subclasses with a mean Pearson correlation coefficient higher than 0.49 to each other were joined into a broader class defined by common marker genes.”). However, we did not merge further into a single *ADARB2*⁺ group because the correlation between CCK and CCK/VIP classes was weak.

Reviewer 2: The *TAC3* population is subdivided in *TAC3* and *TAC3/Sema3A*. But in the discussion, it is mentioned that *TAC3* is also expressed by the CCK/VIP class. Why was it not considered as a separate subclass?

Answer #19:

We did not find a separate cluster expressing CCK/VIP and *TAC3*; rather, *TAC3* was sparsely expressed across the CCK/VIP population (Figure 1). We do comment that *TAC3* is not the best marker gene for the *TAC3* population as explained in previous answers (“Interestingly, we also found a smattering of *TAC3* expression in the CCK and CCK/VIP populations, therefore the *TAC3* population is best defined by its high expression level of *PTPRK* (protein tyrosine phosphatase receptor type K).”)

Reviewer 2: Or why not considering grouping all *TAC3* cells together?

Answer #20:

As already explained previously, the grouping of subclasses into classes was a data-driven process, we did not choose which groups to merge, but rather a correlation threshold. As described in the methods section, “The interneuron subclasses were merged into broader classes based on their correlation. All subclasses with a mean Pearson correlation coefficient

higher than 0.49 to each other were joined into a broader class defined by common marker genes.” We have now added a sentence in the results section to clarify this point:

“We clustered all the interneurons, resulting in 14 clusters which we identified as 14 different interneuron subclasses based on the expression of unique transcriptomic patterns. Merging highly correlated classes (see Methods), we produced a broader classification with eight main classes, which we named after selected marker genes:[...]”

This merging process did not join the CCK/VIP and the TAC3 groups because they were not similar enough considering their whole transcriptomic profile (it can be seen that they are far apart in the dendrogram in Figure 1F), thus we could not group all the *TAC3+* cells together.

Reviewer 2: Some results presented here seem to contradict some pre-existing studies performed in primates. Here are a few examples:

Answer #21:

Indeed, we do aim to harmonize and improve previous classifications performed using snRNA-seq data from human and primate samples as mentioned in our previous answer. Previous studies provide conflicting results among themselves and here we attempt to provide a consensus classification by integrating the largest dataset to date. As mentioned in the answer to the first comment (#15), we have extended our commentary on these apparently conflicting studies in the Introduction and Discussion sections.

Reviewer 2: Previous studies report a much larger population of PV+ interneurons in the striatum and strong overlap with PTHLH.

Answer #22:

There is no previous evidence of a large population of *PTHLH+/PVALB+* interneurons in the human striatum. None of the works from which we sourced human data shows this, but the opposite: *PVALB* expression is low and sparse in all of them.

Krienen *et al.* show a large cluster labeled *PTHLH+/PVALB+* in marmoset data integrated with mouse data (ref. 35, Krienen *et al.*, 2020 Figure 4). The expression of *PVALB* is shown to be far greater in mouse than in marmoset, which makes a small contribution to the cluster. *PTHLH* expression is not shown. The classification of human striatal interneurons in the same paper (Krienen *et al.*, supplementary figure 10) does not show any cluster labeled *PVALB+*, and *PVALB* expression is not shown.

The online tool from Krienen *et al.* (<http://interneuron.mccarrolllab.org>) shows *PVALB* expression in a group labeled *PV+* and another labeled *TH+*. Of these, only *TH+* is shown in the paper, and it is a small cluster.

Lee *et al.* named a relatively large population *PVALB/TH*, but this was clearly not characterized by the expression of *PVALB*. *PVALB* expression in humans is not shown in the original publication, and the dataset shows that *PVALB* has a low, very sparse expression among the three groups of nuclei labeled as interneurons by the authors:

The same lack of *PVALB* is seen in the nuclei not labeled as interneurons by the authors but which do express GABAergic markers:

The work from Tran *et al.* (ref. 36) state that they were unable to observe robust *PVALB* expression on any group of interneurons, and only hypothesized that two of the five interneuron types they describe “likely represent unique PV-expressing interneuron classes” due to the expression of *KIT*, *PTHLH*, and *GADI*.

We presented the data from these authors, and it can be seen that the expression of *PVALB* in both human datasets from Krienen *et al.* and in the other two (Lee *et al.* and Tran *et al.*) is sparse (Figure 6, Supplementary Figure 10).

A recent neuroanatomical study states that the PV+ population constitutes less than 2% of the total striatal interneurons, peaking at 5.4% in the postcommissural putamen (Lecumberri, A., et al. "Neuronal density and proportion of interneurons in the associative, sensorimotor and limbic human striatum." *Brain Structure and Function* 223 (2018): 1615-1625), confirming that PV+ interneurons are actually a small population.

Reviewer 2: Several studies including ones performed in primates, describe a population of THINs. The discrepancies (as well as others) between these studies and previous ones should be better discussed.

Answer #23:

Regarding the previous studies, the previous snRNA-seq works performed in humans have not presented evidence of a population of striatal interneurons characterized by TH expression so far:

Although Krienen *et al.* named a group of human interneurons as *TH+*, there is no *TH* expression in their human dataset nor *TH* expression in human in the online tool they provided to explore their data (<http://interneuron.mccarrolllab.org>), which explicitly states “Not Detected In Interneurons” on the human dataset:

Tran *et al.* does not mention *TH+* interneurons, and Lee *et al.* names a group “*PVALB/TH*”, although the data shows that this group (labeled PV_Interneuron in their public data) only has extremely sparse *TH* expression:

Recent neuroanatomical studies have shown that the TH⁺ expressing cells of the human striatum constitutes less than 1% of the total interneurons, with no striatal area presenting more than 0.7% (Lecumberri, A., et al. "Neuronal density and proportion of interneurons in the associative, sensorimotor and limbic human striatum." *Brain Structure and Function* 223 (2018): 1615-1625).

Aligned with these studies, we hardly found TH-expressing interneurons as we already described in the Results section. Rather than identifying a distinct TH-expressing interneuron population, we established notable genetic parallels between the mouse Th interneuron population and the human TAC3 population that we delineate in our study, asserting their equivalence (see Figure 5). In order to clarify TH expression in the human striatum we have now performed high-sensitive FISH and show a high overlap of TH expression with the MSN marker *DRD1* (Supplementary Figure 9), a fact that is also supported by other works that we already cited in the previous version of our manuscript and now also show in Supplementary Figure 9.

In summary:

- previous human snRNA-seq works with striatal interneurons hardly find any TH expression, and if they define a TH population as Krienen *et al*, it does not present TH expression and differs from the TAC3 population they define.
- in our work we find sparse and low TH expression among the interneurons (see added barplots in Figure 1E and the new Supplementary figure 3) that is in agreement with the low percentage of TH interneurons found in the anatomical studies and the snRNA-seq studies (mentioned above). We find no cluster characterized by TH expression. We do find, as stated in the previous version

of the manuscript, *TH* expression among the MSNs, observation supported by others works cited in ref. 43, 69 and 70 (Saunders, A. *et al. Cell*, 2018; Mao, M., Nair, *et al.* 2019; Darmopil, S. *et al. Eur J Neurosci* **27**, 580–592 (2008) respectively) and that we have also shown from other published striatal snRNA-seq datasets, cited in ref 35-37 (Krienen *et al.* 2020, Tran *et al.* 2021 and Lee *et al.* 2020 respectively), now added in Supplementary figure 9. And in order to further prove this observation we have performed high sensitive FISH using *TH* and the MSN marker *DRD1* (see new Supplementary figure 9).

In response to the reviewer's suggestions, besides the additional mentioned figures we have further emphasized these findings in both the Results and Discussion sections, providing additional clarity and detail (pages 5)

Reviewer 2: On the same lines, the author should acknowledge much more other studies that have investigated different cell types in the striatum both in rodents as well as in primates. While some of these were obtained using different techniques, they provided very important information regarding the classification of GABAergic interneurons, their role in striatal circuits and in behavior. These past studies are cited (for a minority of them) but not used in the context of the classification of different populations of interneurons, the differences observed between studies, the potential implications...

Answer #24: We appreciate the reviewer's insightful comment and acknowledge that the significance of contributions from other works may not have been adequately highlighted in the previous version. We have now (1) added additional references to other previous anatomical studies Kawaguchi Y *et al.* 1995 (ref. 16), Cicchetti F *et al.* 2000 (ref. 17), Petryszyn *et al.* 2018 (ref. 19), Cossette *et al.* 2003, Lallani S.B. *et al.* 2019 (red. 33), Araújo de Góis *et al.* 2023 (ref. 34).); (2) depicted in Figure 1E and Supplementary figure 3 the correspondence between our classification and the classical markers; and (3) extended comments of the previous works in the Results and Discussion sections.

Reviewer 2: The validation of the proposed interneuron classification using FISH is too superficial and should be extended to include more markers. Further, it should be discussed whether the quantifications obtained with both techniques reach similar results. Finally, this has been done mostly for established markers such as SST, NPY, PV. Importantly, FISH of less-established markers such as DACH1 seems to give contradictory results with the snRNA-seq, which brings several questions regarding the validity of the classification.

Answer #25:

We thank the reviewer for their valid comment and have made substantial efforts to address this concern. We have now strengthened our findings by validating our transcriptomic dataset with another independent method. We have performed a high-plex *in situ* platform with subcellular resolution (Xenium, 10x Genomics) to comprehensively characterize the RNA expression profile in tissue sections together with high sensitivity *in situ* hybridization in a subset of up to six donors including both sexes. We have robustly confirmed all 14 subclasses presented in this new taxonomy with a strong agreement between the different approaches (see Figure 3 and Discussion Section). We have also dedicated significant effort and resources to extend the FISH validation we previously had to several interneuron classes and subclasses (see Figure 3F and Supplementary Figures 5 and 9).

Reviewer 2: The number of groups defined in different parts of the study is a bit confusing. The main claim is that there could be 14 subtypes, but when investigating the regional differences this was done only in 8 populations. Then, when looking at functionally relevant genes this was observed for 7 populations (CCK, CCK/VIP, PV, SST/GRIK3, PTHLH, ChAT, TAC3). Finally, when including data from other studies, they found 16 groups. This lack of consistency makes the proposed subdivision more confusing.

Answer #26:

We understand the reviewer's concerns and answer them as follows:

Despite the size of the dataset, the sampling of the less frequent interneuron subtypes was still very limited due to interneurons' general scarcity (Figure 1). Thus, in order to enhance the robustness of our results we analyzed the regional differences at the interneuron class level rather than the subclass level. We have modified the text in the results section to make this more explicit:

“We conducted a comparison of the two striatal regions at the interneuron class level. We chose to focus on classes rather than subclasses to increase the robustness of our results, as the low numbers of some of the subclasses would limit the reliability of the analysis.”

When looking at functional relevant genes we find the 14 subclasses described in this work (see Figure 5A, Supplementary Figure 8), not only those pointed out by the reviewer. We therefore politely disagree with this comment.

When we integrated and clustered data from other studies with our own, we did find that 15 out of the 16 clusters had a clear correspondence to the 14 subclasses found in our dataset, with clusters 1 and 2 both relating to the PTHLH subtype (Figure 6C). The only cluster that we could not map directly to our labels contained mostly (83.6%) cells from the DropSeq dataset of Krienen *et al.* and expressed TAC3 subtype markers. As shown in Figure 6D and in Supplementary Figure 10, the expression pattern of marker genes seems to indicate that these are still interneurons of the TAC3 subtype. This type of variation is common when integrating data from different single cell approaches (in this case DropSeq and 10x Genomics). Thus, we must emphasize the near-perfect correspondence we achieved here. We have added a sentence to remark that we were able to map each cluster of the integrated data with one of our 14 interneuron subclasses:

“Thus, we could identify each of the clusters on the integrated data as one of the 14 interneuron subclasses in our proposed taxonomy.”

Reviewer 2: The process to select interneurons should be better described.

o It says that MSNs were discarded based on D1 or D2 R expression. Are the authors certain that this will not exclude some interneuron populations?

Answer #27:

We appreciate the reviewer's comment and we have now added additional information regarding this in the Methods section to clarify that we did conduct another quality control process after separating the MSNs and the interneuron clusters to ensure we did not exclude any interneurons:

“Nuclei labeled as interneurons were projected onto the first 20 PCs calculated on their 1,500 most variable genes and re-clustered using the Louvain algorithm. The function rank_genes_groups from scanpy⁶³ was used to perform a differential expression analysis between the clusters through a Wilcoxon rank-sum test. We filtered out 2,362 nuclei which formed clusters characterized by low quality control metrics, excitatory markers (*RORB*) or

MSN markers (*PPP1R1B*, *DRD1*, *DRD2*, *MEIS2*), obtaining a final ensemble of 19,339 high-quality interneuron nuclei.”

Reviewer 2: Excitatory neurons were excluded. If samples were taken from the striatum, what are the sources of excitatory neurons?

Answer #28:

Certainly, we implemented stringent measures to preclude potential contaminations from regions other than the striatum, as outlined in the methods section “To remove possible contamination from the claustrum or the amygdala, we removed cells expressing regional markers obtained from the Allen Brain atlas: *NEUROD2*, *TMEM155*, *CARTPT*, *SLC17A7*.” In addition to this, during the classification, we also considered that any cluster characterized by high *RORB* expression could also be a contamination artifact and therefore we decided to remove them. This comprised merely 1,213 cells, representing only 0.67% of the total neurons, with the majority deriving from one donor/sample. This prevalence from a limited subset strongly reinforces the assertion that these cells likely originated from specific dissections.

Reviewer 2: These are selected based on the expression of GAD1 and/or GAD2. Wouldn't this exclude the majority of ChAT+ interneurons?

Answer #29:

We thank the reviewer for this observation. Indeed, although *CHAT* cells also express *GAD1* (but not *GAD2*), we also used *CHAT* expression to ensure we did not discard them. We have now corrected that in the methods section to acknowledge this fact:

“The clusters expressing the inhibitory markers GAD1 and/or GAD2 and/or CHAT and not expressing MSN (PPP1R1B, DRD1, DRD2, MEIS2) or excitatory markers (RORB) were labeled as interneurons.”

Reviewer 2: One more general comment is the definition of a cell type. As mentioned here in lines 423-427: “These two classes, PTHLH and PVALB, do not appear close in their molecular identities in the human striatum when applying unbiased hierarchical clustering; however, when we performed a hypothesis-driven analysis of our data, restricted to relevant genes for neuronal functions such as neurotransmitter receptors or ion channels, these two classes showed a very strong correlation.”

Depending on the analysis methods (or different studies using similar approaches) “cell types” can be considered as unique or different especially considering some gradient expression of some genes. Further, as mentioned above this study doesn't acknowledge enough other criteria (at least as important) used to define a cell type (anatomy, connectivity, electrophysiological properties...) and the research that has been done (in rodents, monkeys, humans...) to classify these cell types, the differences with the current study and potential hypotheses.

Answer #30:

We appreciate the reviewer's comment, and in response, we have revised the text to provide further clarification regarding the specific point we aimed to emphasize (page 21, text in blue). The focus lies on the ongoing discourse within the field concerning the definition of a cell type and its relevance to our data. To enhance this discussion, we have incorporated a recent publication addressing the current state of affairs in ref. 65 (Zheng H et al, *What is a cell type and how to define it? Cell*, 2022).

Reviewer 2: This last fact is also reflected in some comments in the introduction:

- Line 56: “The aspiny striatal interneurons”. Several studies have shown that several populations of interneurons are sparsely spiny.

Answer #31:

We thank the reviewer for this comment, and we have now removed the word “aspiny”.

Reviewer 2: Line 57: “small group of cholinergic giant neurons”. In the majority of studies, cholinergic interneurons are not described as a minor group representing 1-2% of striatal neurons which is more important than several populations of GABAergic interneurons.

Answer #32:

We thank the reviewer for this observation, and we have now removed the word “small”.

Reviewer 2: Lines 59-67: “Since the striatal interneurons have received little attention compared to the MSNs, consensus regarding the populations comprising these neuronal groups and how to identify them is lacking. However, recent advances such as new transgenic reporter mice that target the complete striatal and cortical interneuron repertoire^{15,16}, and single cell/nucleus RNA-sequencing (sc/nRNA-seq) have enabled large-scale approaches to investigate cell diversity based on the individual cell transcriptome^{17,18,19} in different mouse brain areas including the striatum ^{20,21,22}. Using these methods, a recent study identified seven interneuron populations in the mouse striatum based on their molecular and electrophysiological profile: Npy/Sst, Npy/Mia, Cck/Vip, Cck, Chat, Th, and Pthlh ²⁰”
What about all the other evidence obtained before and after the introduction of transgenic mice using methods different than sc/snRNAseq? All the original studies describing the existence of SST, NPY, NGFs, THINs, PV+-FSIs, 5HT3a... are not cited and not accounted for.

Answer #33:

We appreciate this reviewer’s comment and have now added additional references, comments and even new figures (Figure 1E and Supplementary Figure 3), as explained before, to properly acknowledge previous works.

Reviewer 2: Lines 74-79: Most of the studies are limited by the technical approach because they have relied on the classical markers to identify interneuron populations and focused primarily on the cholinergic cells, expressing choline acetyltransferase (ChAT)^{24,25,26}. Prior snRNA-seq studies on the human and non-human primate striatum have highlighted different aspects, such as broad differences across species and brain areas^{27,28} or in health vs. disease²⁹, but lack sufficient interneuron sampling to characterize striatal interneuron diversity.”

This is not true on several levels. 1) Classification of interneurons did not start with snRNAseq and 2) multiple studies have investigated interneuron diversity without only focusing on cholinergic interneurons. These are just not cited here.

Answer #34:

We thank the reviewer's comment and, as explained before, we have now acknowledged other previous works and removed the sentence “and focused primarily on the cholinergic cells”.

REVIEWERS' COMMENTS

Reviewer #1 (Remarks to the Author):

The authors have made a solid effort to incorporate my suggestions. The manuscript is now clearly in continuity with previous research, even though the methodology compared with 'classical' neuroanatomical studies is entirely different. I have just a few minor comments to improve the understandability of the manuscript:

- 1) Supporting Table 8 is much appreciated. It is important to have a table with all abbreviations. However, when these abbreviations are mentioned in the text, there is inconsistency in their spelling: some of them are spelt out the first time they are mentioned, and some of them are not (for example, PTHLH is spelt out on p. 4, after it has been mentioned a few times). Please be consistent.
- 2) In p. 5 (lines 183-5): "The low abundance of TH+ interneurons (<1%) found on Nissl or other stainings is also in line with these observations": Nissl staining is an unspecific labelling, which cannot be used to detect TH neurons. I recommend: "... TH+ interneurons (<1%) found with immunohistochemical techniques is also in line..."
- 3) The difference in cholinergic interneuron measures in the different techniques (snRNA-seq vs spatial transcriptomics) is very remarkable (0.3 vs 13.3%). I may have missed it, but I think that the authors do not discuss this difference. Which number is more reliable to have an idea of the abundance of cholinergic interneurons in the human striatum?
- 4) In their exhaustive analyses of the data, the authors include regional differences between the CN and Put. This is very interesting. However, we should remember that they have analysed just one minor part of the striatum in the anteroposterior axis (the level corresponding to the nucleus accumbens), and they cannot distinguish between dorsal and ventral, medial and lateral aspects. I invite the authors to include this in the limitations of the study. Just a few anatomical studies have analysed the distribution of interneurons in the whole extension of the human striatum, but they found key differences for most subgroups across the anteroposterior, mediolateral and ventrodorsal axes of CN and Put.
- 5) I detected a couple of typos (there may be more):
 - a. In the Abstract: untangle \diamond untangling
 - b. In line 813: coda \diamond code

I have to honestly congratulate the authors on their spectacular contribution and the hard work behind it.

Reviewer #2 (Remarks to the Author):

This revised manuscript from Garma et al., investigates the diversity of striatal interneurons in the human brain. They used snRNA-seq and spatial transcriptomics and propose a new taxonomy of striatal interneurons based on main gene expression.

In this revision, the authors have put immense efforts to address all my previous comments.

These mainly concerned methods clarifications, acknowledgement of previous literature, reasons for potential discrepancies and increasing FISH validations.

They have satisfactorily addressed all these comments.

I have just a minor comment/observation regarding the omission of striatal neurogliaform as well as potential 5HT3a-expressing interneurons in the introduction and whether these interneurons may fall into specific categories in their new classification.

We remark again our gratitude to the reviewers for their thoughtful feedback and constructive suggestions.

Reviewer #1 (Remarks to the Author):

The authors have made a solid effort to incorporate my suggestions. The manuscript is now clearly in continuity with previous research, even though the methodology compared with 'classical' neuroanatomical studies is entirely different. I have just a few minor comments to improve the understandability of the manuscript:

1) Supporting Table 8 is much appreciated. It is important to have a table with all abbreviations. However, when these abbreviations are mentioned in the text, there is inconsistency in their spelling: some of them are spelt out the first time they are mentioned, and some of them are not (for example, PTHLH is spelt out on p. 4, after it has been mentioned a few times). Please be consistent.

Answer 1:

We thank the reviewer for pointing out our mistake about *PTHLH* and revised the manuscript concerning the consistency of introducing abbreviations that are repetitively used and discussed in the manuscript. In the interest of the readability of the text we did not write down the full gene names for genes that are just mentioned as markers or part of a signalling pathway, however the full name of these genes can still be looked up in Table 8.

2) In p. 5 (lines 183-5): "The low abundance of TH+ interneurons (<1%) found on Nissl or other stainings is also in line with these observations": Nissl staining is an unspecific labelling, which cannot be used to detect TH neurons. I recommend: "... TH+ interneurons (<1%) found with immunohistochemical techniques is also in line..."

Answer 2: We thank the reviewer for this observation, and we have now modified the sentence accordingly.

3) The difference in cholinergic interneuron measures in the different techniques (snRNA-seq vs spatial transcriptomics) is very remarkable (0.3 vs 13.3%). I may have missed it, but I think that the authors do not discuss this difference. Which number is more reliable to have an idea of the abundance of cholinergic interneurons in the human striatum?

Answer 3: We thank the reviewer for this comment. We have now included a sentence in the discussion section regarding that: "The proportion of CHAT cells we found in our snRNA-seq data was lower than anticipated based on other studies (0.3% versus 11%^{33,35}). However, our spatial transcriptomic analysis results closely matched the previous data (13.3%), suggesting a technique-specific bias in the snRNA-seq data".

4) In their exhaustive analyses of the data, the authors include regional differences between the CN and Put. This is very interesting. However, we should remember that they have analysed just one minor part of the striatum in the anteroposterior axis (the level corresponding to the nucleus accumbens), and they cannot distinguish between dorsal and ventral, medial and lateral aspects. I invite the authors to include this in the limitations of the study. Just a few anatomical studies have analysed the distribution of interneurons in the whole extension of the human striatum, but they found key differences for most subgroups across the anteroposterior, mediolateral and ventrodorsal axes of CN and Put.

Answer 4: We thank the reviewer for this observation. We have now added a sentence in the discussion section regarding this limitation of our study: “Importantly, these results pertain to a specific region of CN and Pu (see Methods). Slight differences along the anterior-posterior and dorso-ventral axes are expected, as previously described”.

5) I detected a couple of typos (there may be more):

a. In the Abstract: untangle ∅ unangling

b. In line 813: coda ∅ code

I have to honestly congratulate the authors on their spectacular contribution and the hard work behind it.

Answer 5: We thank the reviewer for noting the typos and have now corrected them and revised the entire manuscript.

Reviewer #2 (Remarks to the Author):

This revised manuscript from Garma et al., investigates the diversity of striatal interneurons in the human brain. They used snRNA-seq and spatial transcriptomics and propose a new taxonomy of striatal interneurons based on main gene expression. In this revision, the authors have put immense efforts to address all my previous comments. These mainly concerned methods clarifications, acknowledgement of previous literature, reasons for potential discrepancies and increasing FISH validations. They have satisfactorily addressed all these comments. I have just a minor comment/observation regarding the omission of striatal neurogliaform as well as potential 5HT3a-expressing interneurons in the introduction and whether these interneurons may fall into specific categories in their new classification.

Answer 6: We thank the reviewer for their comment. We have now referred to the NGC and Htr3a-expressing neurons in the introduction. We have included *HTR3a* expression in Figure 5B, as it aligns with the purpose of the analysis. Htr3a expression is also shared by the Th population in mice and TAC3 population in human. A sentence has been added in the Discussion section regarding this: “They also share the expression of the serotonergic receptor HTR3A/Htr3a, which has been used in the cortex as a developmental marker for CGE-derived cells (REF <https://www.ncbi.nlm.nih.gov/pmc/articles/PMC3556905/>). Interestingly, HTR3A/Htr3a is also expressed by the PTLH and CCK/VIP populations in both species. This might indicate a common developmental origin of the PTHLH, TAC3, and CCK/VIP populations.”

Regarding the correspondence between NGC and our present taxonomy, electrophysiological characterization is necessary for a clear delineation. In a previous study using Patch-seq in the mouse striatum (Muñoz-Manchado et al., 2018) we defined two specific NGC markers,

Mia and *Pnoc*, in addition to the already known expression pattern *Npy+Sst-*. To address the reviewer's question, we examined the main markers defined in the mouse striatum (see plot below). Unfortunately, *MIA* is scarcely expressed and *PNOC* is too sparse to define a specific population within *NPY*-expressing cells, suggesting that other markers likely characterize these cells in the human striatum.